# Improving large-scale snow albedo modeling using a climatology of light-absorbing particle deposition

Manon Gaillard[1,2], Vincent Vionnet[1], Matthieu Lafaysse[3], Marie Dumont[3], and Paul Ginoux[4]

[1]Environment and Climate Change Canada, Meteorological Research Division, Dorval, Canada
[2]Ecole Polytechnique, Palaiseau, FRANCE
[3]Univ. Grenoble Alpes, Université de Toulouse, Météo-France, CNRS, CNRM, Centre d'Etudes de la Neige, 38000 Grenoble, France
[4]NOAA/OAR, Geophysical Fluid Dynamics Laboratory, Princeton, New Jersey, USA

**Correspondence:** Vincent Vionnet (vincent.vionnet@ec.gc.ca)

**Abstract.** Light-absorbing particles (LAPs) deposited at the snow surface significantly reduce its albedo and strongly affect the snow melt dynamics. The explicit simulation of these effects with advanced snow radiative transfer models is generally associated with a large computational cost. Consequently, many albedo schemes used in snowpack models still rely on empirical parameterizations that do not account for the spatial variability of LAP deposition. In this study, a new strategy of intermediate complexity that includes the effects of spatially variable LAP deposition on snow albedo is tested with the snowpack model Crocus. It relies on an optimization of the snow darkening coefficient that controls the evolution of snow albedo in the visible range. Optimized values for multi-year snow albedo simulations with Crocus were generated at ten reference experimental sites spanning a large variety of climates across the world. A regression was then established between these optimal values and climatological deposition of LAP on snow at the location of the experimental sites extracted from a global climatology developed in this study. This regression was finally combined with the global climatology to obtain an LAP-informed and spatially variable darkening coefficient for the Crocus albedo parameterization. The revised coefficient improved snow albedo simulations at the ten experimental sites (average reduction in root-mean-square error, RMSE, of 10%) with the largest improvements found for the sites in the Arctic (RMSE reduced by 25%). The uncertainties in the values of the snow darkening coefficient resulting from the inter-annual variability of LAP deposition on snow were computed. This methodology can be applied to other land surface models using the global climatology of LAP deposition on snow developed for this study.

## 1 Introduction

Snow is a key component of the Earth surface energy balance and water cycle (Armstrong and Brun, 2009; Flanner et al., 2011) and provides critical water resources for ecosystems and industrial applications (irrigation, hydro-power, ...) (Sturm et al., 2017; Immerzeel et al., 2020). Snow albedo, the fraction of incident solar radiation reflected by the snow, strongly impacts the surface radiative balance and influences the mass balance of the snow cover through modified snow melt and sublimation (Qu and Hall, 2006; Painter et al., 2017; Skiles et al., 2018; Réveillet et al., 2022). Snow albedo depends on different factors including snow physical properties (e.g. grain size), solar conditions (e.g. solar zenith angle, presence of clouds), local topography, and the

abundance and optical properties of light absorbing particles (Warren and Wiscombe, 1980; Tuzet et al., 2019; He and Flanner, 2020; Picard et al., 2020).

Light-absorbing particles (LAPs) are small impurities often deposited from the atmosphere such as mineral dust (Painter et al., 2010) and black carbon (Flanner et al., 2007), but can also be living organisms such as algae (Cook et al., 2017). Black carbon (BC) is the optically absorbing portion of soot and originates from the incomplete combustion of fossil and biofuels (Bond et al., 2013). Mineral dust originates from arid and semi-arid landscapes and can be transported by the wind over long distances to other parts of the globe (e.g. Di Mauro et al., 2015). When LAPs are present in snow, they darken the snow surface

and decrease its albedo in the visible range (e.g., Warren and Wiscombe, 1980). The broadband albedo of fresh snow can drop from 0.9 to 0.6 due to LAP contamination (Skiles et al., 2018). This direct effect can lead to a faster melting and therefore an increase of snow grain size (through accelerated snow metamorphism), which results in a decrease in snow albedo in the near-infrared range and further increases radiative forcing from the LAPs present in the snowpack. There are therefore two main effects of LAP presence in snow: a direct effect through darkening of the surface, and an associated feedback through

grain-coarsening (Painter et al., 2007; Skiles et al., 2018).

Snow radiative transfer models of various complexity have been developed to simulate the impact of LAPs on snow albedo (Warren and Wiscombe, 1980; Flanner and Zender, 2005; Libois et al., 2013; He and Flanner, 2020; He, 2022). They simulate snow albedo in the visible and near-infrared wavelengths for given snowpack optical properties (specific surface area or optical grain size) and LAP contents. Radiative transfer models have been coupled to snowpack models to simulate the temporal and

spatial evolution of snow radiative properties and account for the albedo feedbacks (Flanner et al., 2007; Tuzet et al., 2017). For example, Libois et al. (2015) and Tuzet et al. (2017) have coupled the Two-stream Analytical Radiative TransfEr in Snow (TARTES) radiative transfer scheme (Libois et al., 2013) with the detailed snowpack model Crocus (Vionnet et al., 2012; Lafaysse et al., 2017). Among numerous applications, these advanced coupled models have been recently used to (i) quantify the impact of LAPs on snow cover evolution in mountainous terrain (Réveillet et al., 2022), (ii) study the impacts of snow

cover on energy fluxes over the Tibetan Plateau (Hao et al., 2023), and (iii) assess the influence of LAPs on snowpack stability (Dick et al., 2023). These coupled models are also used in the context of land and climate models such as in the Community Land Model (CLM Lawrence et al., 2019) and in the land component of the Energy Exascale Earth System Model (Golaz et al., 2019). He et al. (2024) have demonstrated how such models can improve global simulations of snowpack evolution with a positive impact on the simulations of near-surface temperature.

Several challenges arise when combining a snow radiative transfer model with a snow model. They are generally associated with the computational costs and the spectral resolution of the radiative transfer model (Flanner et al., 2007), although methodologies have been developed to optimize spectral snow albedo calculation (van Dalum et al., 2019; Veillon et al., 2021). The need for additional atmospheric forcings (LAP deposition fluxes) to drive the snow model (Tuzet et al., 2017) represents another challenge. For this reason, snowpack models used in land surface and hydrological models still often rely on empirical

parameterizations to describe the temporal evolution of snow albedo (Pedersen and Winther, 2005; Lee et al., 2023). For example, among the 21 snowpack models used in the Earth System Model - Snow Model Intercomparison Project (ESM-Snow MIP, Ménard et al., 2019), only two of them (CLM, (Lawrence et al., 2019) and the Snow Metamorphism and Albedo Pro-

cess model (SMAP Niwano et al., 2012)) explicitly represent the effects of LAP on snow albedo evolution. The other models rely on parameterizations of varying complexity (Supplementary Material of Ménard et al. (2019)) that aim at achieving high computational efficiency with different levels of physical realism.

Simple time/temperature-dependent parameterizations (e.g., Verseghy, 1991; Douville et al., 1995) have been developed to represent the combined effects on snow albedo of multiple snow aging processes (increase in grain size due snow metamorphism, LAP deposition, etc). These parameterizations use fixed time constants that have been optimized using sparse observational data, restricting their extension to untested environments and time periods (e.g. Mölders et al., 2008). More advanced snow models implemented in land surface schemes such as BATS (Dickinson et al., 1993), JULES (Best et al., 2011) and ISBA (Vionnet et al., 2012; Decharme et al., 2016) simulate the snow albedo evolution in different large spectral bands and include explicitly the effect of optical grain size on albedo in these different bands. These models represent only indirectly the impact of LAPs on snow albedo in the visible range through parameters that require calibration. For example, the default version of the Crocus snow scheme used in support of operational avalanche forecasting (Vionnet et al., 2012) relies on a snow darkening coefficient approximately representing the darkening of the snow with time. This parameter was calibrated in the French Alps (Brun et al., 1992) and does not account for the spatial variability of LAP deposition. Applications of the model to other climates such as Antarctica where LAP deposition is extremely low require to tune the rate of albedo decrease in the visible range (Brun et al., 2011). Such tuning is not possible when the model is applied at large scales, which limits the quality of continental-scale snowpack simulations with Crocus (Brun et al., 2013; Mortimer et al., 2020).

The objective of this study is to develop a methodology to improve large scale simulations of snow albedo in snowpack schemes by taking into account the spatial variability of LAP deposition. This methodology is applied in this paper to the default snow albedo parameterization of the detailed snowpack model Crocus to allow a better robustness of the model when applied at large spatial scales. Optimized parameters for snow albedo simulations with Crocus were generated at ten reference experimental sites spanning a large variety of climates. A regression was then established between these optimal parameters and climatological deposition of LAPs (BC and dust) on snow at the location of the experimental sites extracted from a global climatology developed in this study. This regression was finally combined with the global climatology to obtain LAP-informed and spatially-variable parameters for the Crocus albedo parameterization. This methodology could be applied to optimize parameters controlling the albedo evolution in the visible range in other snowpack schemes (e.g., Dickinson et al., 1993; Best et al., 2011; Decharme et al., 2016). The paper is organized as follows. Sect. 2 presents the snow albedo parameterization in Crocus. Sect. 3 then details the model configuration, the evaluation data and method as well as the datasets used to build the global climatology of LAP deposition on snow. Results are presented in Sect. 4 followed by a discussion in Sect. 5. Finally, Sect. 6 summarizes the results and offers concluding remarks.

## 2 Snow albedo parameterization in Crocus

Crocus is a multilayer snowpack model used for avalanche hazard forecasting, climate, and hydrology applications (Brun et al., 1992; Vionnet et al., 2012; Lafaysse et al., 2017). It simulates the seasonal evolution of the physical properties of the

**Table 1.** Equations representing the snow albedo and the snow absorption coefficient for the three spectral bands in Crocus. The parameters are as follows: $d_{opt}$ (m) is the optical grain diameter of the snow, $\rho$ is the snow density, $P$ (Pa) is the mean pressure at the site, $P_{CDP}$ (Pa) is the mean pressure at the Col de Porte site, $A$ (days) is the age of the snow, and $\gamma$ (days) is the snow darkening coefficient. Adapted from Table 4 in Vionnet et al. (2012).

| Spectral band | Spectral albedo $\alpha_{spectral\,band}$ | Absorption coefficient $\beta$ (m$^{-1}$) |
|---|---|---|
| 0.3—0.8 µm | $\alpha_{0.3-0.8\mu m} = \max(0.6, \alpha_i - \Delta\alpha_{age})$ <br> $where: \alpha_i = \min(0.92, 0.96 - 1.58\sqrt{d_{opt}})$ <br> $and: \Delta\alpha_{age} = \min(1, \max(\frac{P}{P_{CDP}}, 0.5)) \times 0.2\frac{A}{\gamma}$ | $\beta_{0.3-0.8\mu m} = \max(40, 0.00192\,\frac{\rho}{\sqrt{d_{opt}}})$ |
| 0.8—1.5 µm | $\alpha_{0.8-1.5\mu m} = \max(0.3, 0.9 - 15.4\sqrt{d_{opt}})$ | $\beta_{0.8-1.5\mu m} = \max(100, 0.01098\,\frac{\rho}{\sqrt{d_{opt}}})$ |
| 1.5—2.8 µm | $\alpha_{1.5-2.8\mu m} = 346.3d' - 32.31\sqrt{d'} + 0.88$ <br> $where: d' = \min(d_{opt}, 0.0023)$ | $\beta_{1.5-2.8\mu m} = +\infty$ |

snowpack and its vertical layering. For each snow layer, Crocus simulates the evolution of the thickness, density, liquid water content, temperature, age, and snow microstructure represented by the snow specific surface area and snow grain sphericity. The sphericity is a semi-empirical variable that describes the ratio of angular versus rounded shape in a given snow layer (Brun et al., 1992; Carmagnola et al., 2014). This study relies on the version of Crocus that has recently been implemented as an additional option for snow simulations in the Soil, Vegetation and Snow version 2 (SVS-2) land surface scheme (Garnaud et al., 2019; Vionnet et al., 2022). SVS-2 is the land surface scheme used at Environment and Climate Change Canada (ECCC) in preparation of the Terrestrial Snow Mass Mission (Derksen et al., 2021). Within SVS-2, Crocus is coupled to a multi-layer soil model including soil freezing (Boone et al., 2000). In this version of Crocus, the maximum number of simulated snow layers was set at 20 to test a viable configuration for an eventual operational implementation covering the whole Canadian territory at 500-m resolution; such a configuration needs to balance accuracy and computational time. The rest of this section describes the default snow albedo parameterization in Crocus.

Snow albedo in Crocus is split between three spectral bands of incoming solar radiation: one in the visible ($0.3 - 0.8$ µm), and two in the near infrared ($0.8 - 1.5$ µm and $1.5 - 2.8$ µm). The albedo in each spectral band is calculated using a different equation (Table 1) as proposed by Brun et al. (1992) based on the work from Warren (1982) and Sergent et al. (1987). In the visible band, snow albedo depends mostly on the amount of LAPs, which is parameterized by the age of snow, and on the snow microstructure, represented by the optical diameter of snow (Carmagnola et al., 2014). In the near-infrared bands, the spectral albedo depends only on the optical diameter of snow. The evolution of the optical diameter in Crocus is computed using metamorphism laws as described in Brun et al. (1992) and Carmagnola et al. (2014). For a given layer, for dry snow, the temporal evolution of the optical grain size is a function of the vertical temperature gradient, whereas, for wet snow, the increase in optical grain size with time depends on the snow liquid water content. New snow in Crocus is characterized by an optical diameter of 0.1 mm (specific area of 65 m$^2$kg$^{-1}$) and results in an increase in snow albedo. The snow albedo in each

band is obtained as a depth-weighted average of the albedo of the two upper snow layers. The properties (optical diameter, $d_{opt}$, age, $A$) of each layer are used to compute their respective albedo. This method is applied to avoid time discontinuities in the simulated snow albedo resulting from layer aggregation.

To obtain the total albedo, the model assumes by default that the incoming shortwave radiation is split into the three bands as follows: 71% in the 0.3—0.8 µm range, 21% in the 0.8—1.5 µm range, and 8% in the 1.5—2.8 µm range. The total albedo is therefore defined as:

$$\alpha = 0.71\alpha_{0.3-0.8\mu m} + 0.21\alpha_{0.8-1.5\mu m} + 0.08\alpha_{1.5-2.8\mu m} \tag{1}$$

Crocus simulates the penetration of incoming shortwave radiation into the snowpack and its absorption assuming an exponential
decay of radiation with increasing snow depth. The absorption coefficient in the different spectral bands (Table 1) depends on the optical diameter and on the density of each snow layer.

    The impact of LAP deposition is parameterized by the age of snow through a snow darkening coefficient, $\gamma$ (Table 1). The snow age in Crocus corresponds to the time in days since the last snowfall for a given snow layer. A new layer made of fresh snow is given a snow age of 0 and the snow age increases with time if no snowfall occurs. The snow darkening coefficient
controls the impact of the snow age $A$ on the temporal evolution of the albedo in the visible band, as shown on Fig. 1. If $\gamma$ is high, the albedo decreases slowly with the snow age, which is characteristic of a snowpack that receives low LAP deposition. In contrast, if $\gamma$ is low, the effect of snow darkening on albedo is larger, so that the snow albedo decreases more quickly, which can be associated with higher LAP concentrations at the snow surface. Fig. 1 illustrates the dependency of the snow albedo to the snow age for different values of $\gamma$ and a constant value of the optical diameter. The default value of $\gamma$ used until now in
Crocus simulations is 60 days. It has been set during the early developments of Crocus at the Col de Porte experimental site in the French Alps (Brun et al., 1992). When applied in Antarctica, Brun et al. (2011) used a value of 900 days to take into account the exceptionally clean atmosphere over the Antarctic plateau so that the snow albedo decrease was drastically reduced in the visible range. The default snow albedo parameterization in Crocus does not include the change in the ratio of incoming radiation in the three bands as a function of the sky conditions and of the solar zenith angle. The impact of these limitations is
discussed in Sect. 5.

    The narrowband albedo in the visible range also includes an empirical term that was initially designed to reflect the impact of elevation on the concentration of LAPs in the snow when applied in the French Alps for avalanche hazard forecasting (Brun et al., 1992). This element is the pressure term, $A_P$ in the last line of the $0.3 - 0.8$ µm albedo equations in Table 1:

$$A_P = \min(1, \max(\frac{P}{P_{CDP}}, 0.5)), \tag{2}$$

where $P_{CDP}$=870 hPa is the mean pressure at the Col de Porte experimental site in the French Alps (1325 m) (Morin et al., 2012). $A_P$ decreases with the site's elevation, thus decreasing the impact of the snow age on spectral albedo and mimicking the effect of lower LAP concentrations at high elevations. The idea behind this mechanism is that sites which are at a high elevation in the French Alps are supposedly further away from pollution sources such as roads and cities located in valleys, and consequently have lower concentrations of LAPs. However, the added value of this parameterization has never been

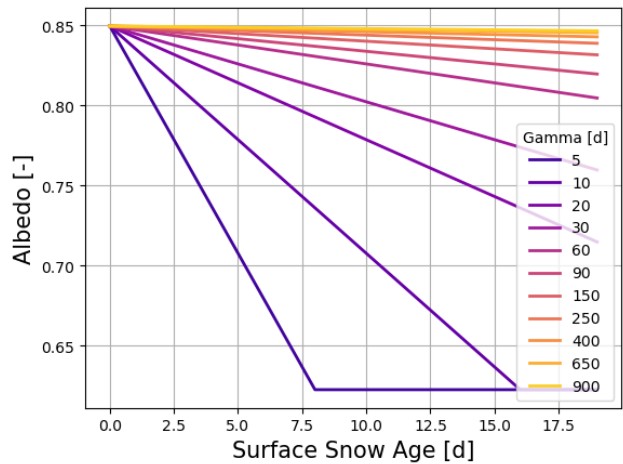

**Figure 1.** Graphic representation of the dependency of the snow albedo in the visible spectral band $(0.3 - 0.8\mu m)$ to the snow age for different values of $\gamma$ and a constant value of the optical diameter, $d_{opt} = \frac{6}{\rho_{ice} SSA}$, where $\rho_{ice} = 917 \text{ kgm}^{-3}$ (at 0°C, Libois et al. (2015)) and $SSA = 50 \text{ m}^2\text{kg}^{-1}$ (maximum for fresh snow at the Dome C site, Libois et al. (2015))

objectively evaluated and the relation between LAP impact and elevation could be more complex, as suggested in Réveillet et al. (2022), where a stronger influence of LAPs on the snow melt onset date was found at higher elevations. In the context of this work, considering large scale applications of Crocus, the pressure term was not considered when computing $\Delta\alpha_{age}$ (Table 1). Instead, the impact of elevation on the deposition of LAPs was included by considering the altitudinal gradient of LAP deposition around each study site as described in Sect. 4.2.2.

## 3   Data and methods

First, the skills of snow albedo simulations with Crocus were examined for different values of the snow darkening coefficient, $\gamma$, at 10 reference sites around the world, to find optimal ranges of $\gamma$ for each site. Then, a global climatology of mean-annual LAP deposition rates on snow was computed, from which the values at each of the 10 reference sites were extracted. A cross analysis was then carried out to find a simple relationship between the LAP climatology and the optimal ranges of $\gamma$ at the reference sites. This relationship was finally applied to the global climatology of LAP depositions, to obtain a global dataset of new and improved $\gamma$ values that can be used for large scale application of Crocus.

### 3.1   Snow albedo simulations

#### 3.1.1   Data

This study used data from 10 sites spanning various snow cover types such as taiga snow, alpine snow and maritime snow (Table 2 and Fig. 2). These sites were selected because of the availability of reference meteorological data to drive snowpack

simulations and reference snow measurements (including snow albedo) to evaluate the simulations. Six of these sites were taken from the ESM-SnowMIP dataset (Ménard et al., 2019), a series of ten sites with standardized and quality-checked observations and meteorological driving data. Among the ten original ESM-SnowMIP sites, the three located in forested areas (Old Aspen, Old Black Spruce, and Old Jack Pine) were not selected due to the additional effect of forest debris on snow albedo (Melloh et al., 2001). Out of the seven remaining sites, snow albedo measurements were not provided for two sites – Reynolds Mountain East (Idaho, USA), and Sodankylä (Finland) – in the original ESM-SnowMIP dataset (Ménard et al., 2019). However, the Sodankylä site has more recent albedo measurements which were not included in the ESM-SnowMIP dataset but were made available by the Finn. Met. Inst. (2018). A corresponding meteorological forcing file was built extending in time the methodology of Essery et al. (2016) as detailed in Sect. S2.1 in Supplementary Material. Therefore, Sodankylä was included with this alternative observation and meteorological dataset.

To include more diverse locations, several other sites were also considered in the analysis. Three sites in the Canadian Arctic region (Bylot, Umiujaq, and Trail Valley Creek) were added, as well as one alpine site in Austria (Kühtai). This yielded a total of 10 sites for this study. To guarantee homogeneity between all ten sites, the observed daily-average albedo at the added sites was computed from hourly incoming and outgoing shortwave radiation using the same methods as for the ESM-SnowMIP sites (Morin et al., 2012; Ménard et al., 2019). Hourly values of incoming and outgoing radiation were filtered to discard hours when there was snowfall, when the incoming radiation was below 20 W m$^{-2}$ (to remove hours when the sun is low on the horizon), or when the outgoing radiation was below 2 W m$^{-2}$. For the five sites which were added in this study (Bylot, Umiujaq, Trail Valley Creek, Kühtai, and Sodankylä), another filter was added ensuring that the hourly incoming radiation was greater than the outgoing radiation. For all sites, the daily-averaged albedo was then computed by dividing the sum of hourly outgoing radiation values by the sum of hourly incoming radiation values. Days with less than 5 hours of valid radiation measurements were discarded. At each site, the atmospheric forcing is representative of the local meteorological conditions and is mostly made of observations (Ménard et al., 2019). Hourly incoming longwave radiation from the ERA5 reanalysis (Hersbach et al., 2020) was used at Bylot to drive Crocus due to uncertainty in the observed longwave radiation at this site (Domine et al., 2021). For all the other sites, the observed longwave radiation was used.

On top of observed albedo data, daily observations of snow depth ($SD$) were also extracted at these ten sites. They were used to select days for the evaluation of albedo simulations (Sect. 3.1.2), and to evaluate the final results of this study (Sect. 3.3).

### 3.1.2 Methods

Multi-year snowpack simulations were carried out with the snowpack model Crocus within SVS2 at the ten experimental sites described in Sect. 3.1.1. The model configurations (meteorological forcing heights, soil texture, vegetation type, etc.) were obtained from the reference papers describing each site (Table 2). A spin-up period of four years at each site (the first four years of forcing dataset) was considered to provide initial conditions for the surface and soil column. The physical options used in the multiphysics version of Crocus are detailed in Table A1.

**Table 2.** List of the meteorological sites used in this study and some of their characteristics.

| Site Name | Code | Source | Time Period | Elevation (m) | Latitude, Longitude (°) | Snow Cover |
|---|---|---|---|---|---|---|
| Bylot | BYL | Domine et al. (2021) | 2014-2019 | 22 | 73.15, -80.00 | Taiga |
| Col de Porte | CDP | Ménard et al. (2019) | 1994-2014 | 1325 | 45.30, 5.77 | Alpine |
| Kühtai | KUT | Krajči et al. (2017), Günther et al. (2019) | 1990-2013 | 1920 | 47.21, 11.01 | Alpine |
| Sapporo | SAP | Ménard et al. (2019) | 2005-2015 | 15 | 43.08, 141.34 | Maritime |
| Senator Beck | SNB | Ménard et al. (2019) | 2005-2015 | 3714 | 37.91, -107.73 | Alpine |
| Sodankylä | SOD | Finn. Met. Inst. (2018), Corcket (2023) | 2012-2022 | 179 | 67.37, 26.63 | Taiga |
| Swamp Angel | SWA | Ménard et al. (2019) | 2005-2015 | 3371 | 37.91, -107.71 | Alpine |
| Trail Valley Creek | TVC | Tutton et al. (2024) | 2013-2018 | 91 | 68.75, -133.50 | Taiga |
| Umiujaq | UMQ | Lackner et al. (2023) | 2012-2020 | 130 | 56.56, -76.48 | Taiga |
| Weissfluhjoch | WFJ | Ménard et al. (2019) | 1996-2016 | 2536 | 46.83, 9.81 | Alpine |

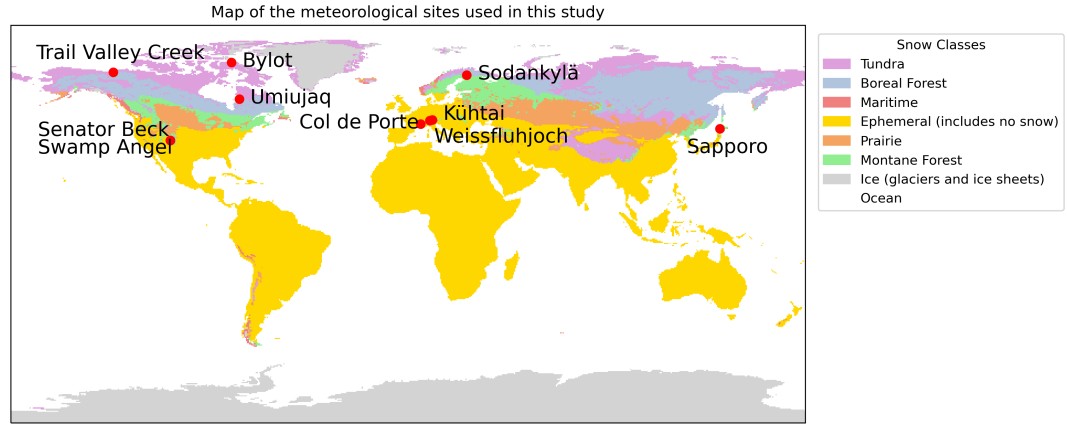

**Figure 2.** Map showing the location of the experimental sites used in this study. The global seasonal snow classification of Sturm and Liston (2021) is used to show the snow classes.

Simulations were first run at each of the ten sites for values of $\gamma$ between 5 and 900 days, with an eleven-point distribution based on an exponential: [5, 10, 20, 30, 60, 90, 150, 250, 400, 650, 900] days. The maximal value of 900 days corresponds to the value used by Brun et al. (2011) in Antarctica. The minimum values of 5 days corresponds to a decrease in snow albedo of 0.2 in one week (Fig. 1), representative of maximal albedo decrease during LAP deposition events (e.g. Dumont et al., 2020). A range of $\gamma$ containing the values which yield the most skilled simulations was found at each site, using a method described further on in this section. After this first round of simulations with $\gamma$ in the [5, 900] days interval, a second round of simulations was run at each site for a zoomed interval around the range of best $\gamma$ values found during the first round. The zoomed intervals were manually chosen for each site as a compromise between granularity of the explored $\gamma$ values and reasonable computing time. As in the first run, the intervals were based on exponentials. This second round yielded more refined ranges of best $\gamma$ values at each site. For the evaluation, the hourly simulated snow albedo was averaged over each day to retain an aggregated daily value similar to the observations.

Two criteria were imposed to select the days used to evaluate the simulated snow albedo. These criteria were all compiled for each site and year into a single mask which was then applied to all observation and simulation series for this site and year. The first criterion ensured that the observed albedo corresponded to an actual snow albedo value. It made sure that the albedo measurement had been collected over a fully snow-covered ground and that the ground surface beneath the snow cover did not affect the measurement. Observed SD had to be higher than 20 cm. For high-latitude sites (above 60°N), this criterion was relaxed to 10 cm because there is less precipitation at these sites, so SD did not reach far over 20 cm for certain years. Albedo simulations with the radiative transfer model SNICAR v3 (Flanner et al., 2021) confirmed the ground surface beneath the snow cover does not modify the snow albedo by more than 2 % for a 10-cm thick snowpack (see Sect. S2.2 of the Supplementary Material). Observed albedo also had to be higher than 0.5, which was a looser version of the criteria used in Lafaysse et al. (2017), where all albedo values below 0.6 were discarded. Finally, days where SD was null in the simulation with the lowest value of $\gamma$ were discarded, keeping only days where there was in fact snow on the ground in all the simulations. This assumption is reasonable because simulated albedo is only computed from the surface layer physical properties so that the simulated albedo is not modified in case of thin simulated snowpacks.

The second criterion was to select periods when the impact of LAPs on snow albedo was significant. These periods physically correspond to days where the snow has aged, accumulating dry depositions of LAPs, and possibly gathering at the surface LAPs from the lower layers if melting occurred (Tuzet et al., 2017). The snow age of the simulation with the highest $\gamma$ had to be higher than a given value $A_{lim}$ (in days). Several criteria guided the choice of $A_{lim}$:

- $A_{lim}$ had to be high enough for the impact of LAPs to be perceptible.

- $A_{lim}$ had to be high enough to discriminate between different values of $\gamma$, as its effect becomes increasingly noticeable with the increase of snow age (Fig. 1).

- $A_{lim}$ had to be high enough to reduce the chance of having snow accumulated on the incoming solar radiation sensors and affecting the quality of the data (Lapo et al., 2015).

- $A_{lim}$ had to be low enough to retain a sufficient number of good quality days.

In fine, the value $A_{lim} = 5$ days was retained as providing a compromise between these criteria.

Two metrics were used to quantify the skills of the snow albedo simulations: the bias and the root mean square error (RMSE).
The bias quantifies the tendency of a simulation to systematically overestimate or underestimate the snow albedo. Its definition is as follows to compare the observed ($obs$) and simulated ($sim$) times series for a given site, year, and value of $\gamma$:

$$Bias_{site,year,\gamma} = \mathbb{E}[sim_{site,year,\gamma}] - \mathbb{E}[obs_{site,year,\gamma}], \tag{3}$$

where $\mathbb{E}[x]$ is the mean of $x$. The RMSE was also used to complement the bias. Its definition is as follows:

$$RMSE_{site,year,\gamma} = \sqrt{\mathbb{E}[(sim_{site,year,\gamma} - obs_{site,year,\gamma})^2]}. \tag{4}$$

For each site, and for each simulation with a given value of $\gamma$, the scores were computed for each year. A group of yearly scores for each value of $\gamma$ was obtained at each site. The median over the years was computed to derive a single value of the score for each $\gamma$ at the site.

Albedo measurements are subject to instrumental uncertainties in both intrinsic precision and snow accumulation on incoming radiation receptors (e.g. Lejeune et al. (2019); Lapo et al. (2015)). This means that there was a need to account for observation uncertainty when computing the error metrics, as two slightly different scores for two values of $\gamma$ could turn out to be statistically equivalent. In such a case, a range of best $\gamma$ values would need to be considered instead of a single best $\gamma$ value at each site. This range would contain the value with the best score at the given site, as well as all the other values which are statistically equivalent to it. Taking inspiration from Lafaysse et al. (2017), and focusing on the RMSE score, a statistical method described in Appendix B was used to find the range of best $\gamma$ values at each site.

## 3.2 Climatology of light-absorbing particles deposition on snow

The goal of this section is to obtain a global climatology of daily LAP deposition rates on snow. Due to the important variability of the length of the snow season around the world, global snow data was needed to select only deposition over snow. A climatology of daily LAP deposition data was therefore combined with snow cover data in order to retain only LAP depositions on snow-covered ground.

### 3.2.1 Data

The LAP deposition data was taken from Zhao et al. (2018). This dataset has been generated with the NOAA/OAR Geophysical Fluid Dynamics Laboratory Coupled Model version 4 (CM4; Zhao et al. (2018)) running for 37 years (1979-2015) on a C192 grid (192 grid points on each edge of the cube projected on Earth). GFDL-CM4 is driven by IPCC CMIP6 forcings, including the emissions of BC (fossil fuel burning, aircraft, shipping, and biomass burning). Dust emissions are calculated on-line with a cubic dependency on surface winds using a fixed dust source function, and a constant threshold of wind erosion (Ginoux et al., 2001). The aerosols are removed from the atmosphere by in- and below-cloud deposition, dry deposition at the surface, and gravitational settling for dust particles. The model results are interpolated from C192 to a Cartesian latitude-longitude grid

of 0.5 by 0.5 degrees resolution. The dataset consists of different percentiles (1, 5, 10, 25, 50, 75, 90, 95 and 99) of BC and dust deposition rates (in kg m$^{-2}$ s$^{-1}$) over the period 1979-2015, for each day of the year (see an example on Fig. S1 in the Supplementary Material). The deposition rates for each type of LAP correspond to the total deposition rates (including both wet and dry depositions). The deposition fluxes at the surface are given for an elevation corresponding to the ground elevation plus half of the thickness of the first atmospheric layer ($\sim$ 15 m). This elevation is referred to as the elevation of the GFDL grid in the rest of the paper. This dataset has been described and evaluated in Zhao et al. (2018). A complementary analysis of the accuracy of the GFDL dataset is presented in Sect. S1.2 of the Supplementary Material. In particular, it shows that the spatial variability of the mean annual dust deposition is well captured by the GFDL dataset (Fig. S3 in the Supplementary Material).

For the daily snow cover data, the Global Multisensor Automated Snow and Ice Mapping System (GMASI) product (Romanov, 2017) was used. Based on a combination of satellite observations in the visible, infrared, and microwave bands, it provides a daily partition of surface cover between water, bare ground, ice-covered water, and snow-covered ground, at a global scale from 1988 to the present (see an example on Fig. S2 in the Supplementary Material). This data is discretized along a 0.04° by 0.04° longitude-latitude grid, which is equivalent to about 4 kilometers by 4 kilometers at the equator. There are two versions of GMASI available: version 3 runs from 1988 to 2018, while version 4 is available from 2006 to 2023. In this study, the earlier version 3 was used up until 2005, after which it was replaced by the current version 4. The authors of the GMASI product made the choice to assume the permanent absence of snow between -25°N and +25°N longitudes, except in South America where this is only true East of -60°E to capture the presence of snow in the Andes. This means that the climatology produced in this study has the same characteristics. The accuracy of the GMASI snow product has been evaluated by Romanov (2017) versus in situ and other remote sensing datasets. Over the continental United States, he found that the percentage of correct snow identifications when compared with in situ observations ranged between 75 and 85 % for the winter months and 94 % over the full year. When compared with the Interactive Multisensor Snow and Ice Mapping System (Helfrich et al., 2007) over the whole land area of the Northern Hemisphere, the agreement remained above 90 % all year long. The largest differences between the two products were found during the fall and the spring due to fast changes in the snow cover.

### 3.2.2 Methods

The climatology of dust and BC deposition over snow-covered ground was computed using the GFDL and GMASI datasets. Proportions of snow-covered ground for each day of the year were first computed over the GMASI grid by averaging the GMASI product over the period 1988-2015. A weighted average was then computed to obtain the mean daily LAP deposition over snow-covered ground: for each day the median deposition rates were first multiplied by the corresponding proportion of snow-covered ground, and these products were summed over the full year. Then this sum was divided by the sum over the year of all proportions of snow-covered ground. For a given location and for a given type of LAP (BC or mineral dust), the mean daily deposition rate on snow, $D_{\mathrm{mean}}$ (in kg m$^{-2}$ d$^{-1}$), is therefore written as:

$$D_{\mathrm{mean}} = \frac{\sum_{d=1}^{365} \delta_d p_d}{\sum_{d=1}^{365} p_d} \times 60 \times 60 \times 24, \tag{5}$$

where $\delta_d$ is the median deposition for that type of LAP over 1981-2015 on the day of the year $d$ (in kg m$^{-2}$ s$-1$), and $p_d$ is the proportion of snow-covered ground over 1988-2015 for the day of the year $d$. Febuary 29 was ignored for leap years, as if all years over the period had 365 days. Two other limitations can be identified in this calculation: (i) GMASI begins in 1988, while the GFDL dataset accounts for years between 1981 and 1987 and (ii) the GFDL dataset gives the median of the deposition rates and not the mean values.

LAP deposition (dust in particular) on snow is characterized by a strong inter-annual variability which can impact differently the evolution of the snow cover from one year to another (e.g. Réveillet et al., 2022). To quantify this inter-annual variability, the percentiles 25 and 75 for daily dust and BC deposition rates of the GFDL dataset were considered to obtain an estimation of the typical daily average deposition of dust and BC on snow for a low LAP year ($D_{\text{low}}$) and for a high LAP year ($D_{\text{high}}$). The median deposition in Eq. 5 for a given type of LAP was replaced by the corresponding percentile 25 (75) for each day of the year for a low (high) LAP year.

Because of their different optical properties, dust and BC have different radiative impacts on the snowpack for the same deposited mass (Clarke et al., 2004). To present a single climatological value of LAP deposition, it was therefore convenient to express dust deposition in terms of equivalent BC deposition. This corresponds to the BC deposition that would have the same integrated radiative impact as the considered dust deposition, over the studied spectral bands. Tuzet et al. (2019) examined expressions for an equivalent BC concentration in snow $c_{eq,BC}$ integrated over the $0.35 - 0.9$ μm spectral band. An average of the slope in Fig. 1 of Tuzet et al. (2019) yielded the following expression:

$$c_{eq,BC} = c_{BC} + 0.0033 c_{dust}, \tag{6}$$

where $c_{BC}$ is the BC concentration in snow and $c_{dust}$ is the dust concentration, all concentrations being expressed in $ng\ g^{-1}$. This relationship can also be used on deposition rates ($D_{\text{mean}}$, $D_{\text{low}}$ and $D_{\text{high}}$) by linearity. The inter-annual variability of BC, dust and total LAP deposition on snow with respect to the climatology is characterized by a coefficient of variability, $C_{V,D}$:

$$C_{V,D} = \frac{D_{\text{high}} - D_{\text{low}}}{D_{\text{mean}}} \tag{7}$$

From the global climatology, values of dust, BC, and equivalent BC deposition were extracted for each of the experimental sites described in Sect3.1.1. The effect of elevation on the climatological mean BC and dust depositions was also investigated in the vicinity of the five mountain sites: Col de Porte, Kühtai, Senator Beck, Swamp Angel, and Weissfluhjoch. The elevation of the GFDL climatology grid cell closest to each site was first compared to the real elevation at the site. The regional gradient of mean LAP deposition ($D_{\text{mean}}$) with elevation was then determined around each site. It was computed using a linear regression of the total LAP deposition rate as a function of the elevation using the grid cells surrounding each site in a 25-cell by 25-cell square (which was approximately equivalent to a 5-cell by 5-cell square of the GFDL grid). An elevation-corrected total mean LAP deposition rate at each site was then computed using the linear regression and the actual elevation of the site and the corresponding elevation of the GFDL dataset. The addition of half of the first atmospheric layer of the GFDL dataset was neglected here, as it was very small compared to the sites' ground elevations. A regional gradient with elevation at each mountain site was also computed for $D_{\text{low}}$ and $D_{\text{high}}$ and used to correct the corresponding deposition values.

### 3.3 Cross-analysis

The final step of the analysis consisted in identifying a relationship between the range of statically equivalent best $\gamma$ values and the mean LAP deposition rate at each site. A specific linear regression method, described in Appendix C, was proposed to find the optimal relationship between the statically-equivalent range of $\gamma$ and the individual LAP deposition rate.

To assess the added value of this LAP-dependent parameterization of $\gamma$ in large scale simulations compared to the default constant setting, a final evaluation was carried out to quantify the impact of using the new values of $\gamma$ on simulated snow albedo and snow depth at the ten experimental sites. The RMSE was computed to compare simulated and observed albedo and $SD$ at each site and for each year, for two values of $\gamma$: the default value of 60 days, and the new value resulting from the regression proposed in this study. The median of the yearly scores was then considered, as in Sect. 3.2.2. For each year, the RMSE was computed when the ground was fully covered by snow as described in Sect. 3.2.2. The filter on the snow age was not applied. The relative change in RMSE resulting the use of the new value of $\gamma$ was finally computed at each site as:

$$I_v = \frac{\widehat{RMSE}_{v,60} - \widehat{RMSE}_{v,\gamma_R}}{\widehat{RMSE}_{v,60}} \tag{8}$$

where $v$ is the considered variable (snow albedo or $SD$) and $\gamma_R$ is the new value obtained from this study.

Finally, using the optimal regression obtained between the climatology of LAP deposition ($D_{\mathrm{mean}}$) and $\gamma$, a global map of optimal $\gamma_{\mathrm{mean}}$ values was derived from the global map of $D_{\mathrm{mean}}$. This optimal regression was also applied to the LAP deposition rate on snow for a low ($D_{\mathrm{low}}$) and a high LAP year ($D_{\mathrm{high}}$) to quantify the uncertainty in the estimation of $\gamma$ resulting from the inter-annual variability of LAP deposition on snow. Values of $\gamma_{\mathrm{low}}$ and $\gamma_{\mathrm{high}}$ were derived and a coefficient of variability for $\gamma$ was computed. To obtain a positive coefficient, it is $\gamma_{\mathrm{high}}$ which is subtracted to $\gamma_{\mathrm{low}}$, since the values of $\gamma$ for a low LAP year are higher than for a high LAP year.

$$C_{V,\gamma} = \frac{\gamma_{\mathrm{low}} - \gamma_{\mathrm{high}}}{\gamma_{\mathrm{mean}}} \tag{9}$$

## 4 Results

### 4.1 Snow albedo simulations

Fig. 3 shows an example of the seasonal evolution of observed and simulated snow depth and snow albedo for the Col de Porte experimental site during winter 1998-1999. The age of surface snow is also shown for the simulations with different values of $\gamma$. As expected, simulations with low values of $\gamma$ are associated with a faster decrease in snow albedo than simulations with high values of $\gamma$ during all the periods following snowfall that refreshes the snow surface. Snow depth starts diverging between the different simulations in the second half of the snow season once the peak of snow depth has been reached. Snow depth in simulations with low values of $\gamma$ decreases faster due to larger melt rates associated with lower snow albedo and increased snow compaction due to the presence of liquid water in the snowpack. The snow cover totally vanished 13 days later in the

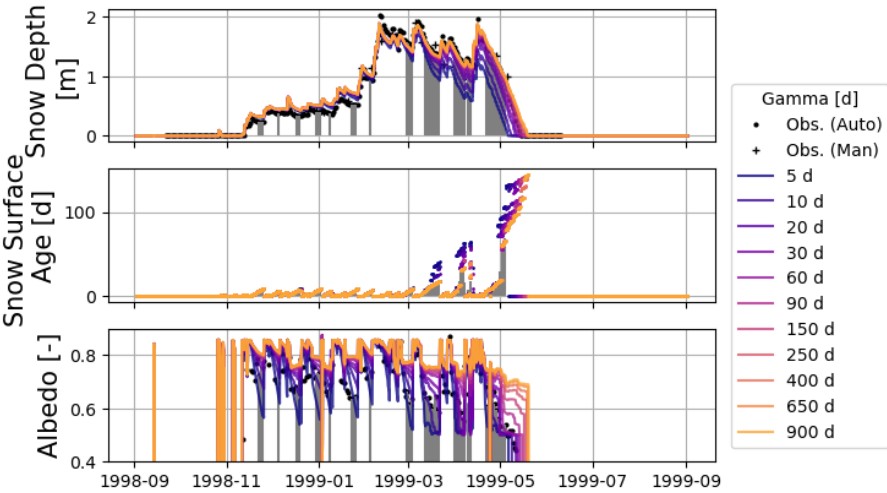

**Figure 3.** Time series of observed (black dots) and simulated (colored lines, one for each value of $\gamma$) values for $SD$ (in meters, top), snow age of the surface layer (in days, middle), and albedo (no unit, bottom) at the Col de Porte site over the 1998 – 1999 snow year. The grey vertical lines mark the days which have been selected for the evaluation. Note that observed data are not available for snow age.

simulation using $\gamma$ = 900 days compared to the simulation with $\gamma$ = 5 days. The vertical grey lines on Fig. 3 show the days selected to compute the albedo evaluation using the criteria listed in Sect. 3.1.2. As designed in Sect. 3.1.2, they cover periods of decreasing albedo, far enough from snowfall events (increases in $SD$). The average number of selected days per snow season was examined at each site and can be found in Supplementary Material (Table S4). On average over all sites and snow seasons, 41 days were retained per season, and all sites but two are grouped around this value. The two outliers are Sapporo with 12 days selected due to its particularly short snow season, and Weissfluhjoch with 73 days.

Error metrics (bias and RMSE) for snow albedo were then computed for each year at each site. Fig. 4 shows an example of the distribution of these error metrics for different values of $\gamma$ at the Col de Porte experimental site. Each boxplot summarizes the distribution of the score for all the available years for a given value of $\gamma$. Results for the two rounds of evaluation considering two ranges of values for $\gamma$ (Sect. 3.1.2) are shown on Fig. 4. At Col de Porte, the optimal values of $\gamma$ ranged from 13 to 20 days: these values were associated with the lowest values of RMSE and had snow albedo biases close to zero. The box plots for all other sites for the second round of evaluation can be seen in Supplementary Material (Fig. S5 to S13).

The resulting optimal ranges of $\gamma$ obtained at each site were compiled in Fig. 5 (logarithmic scale on Fig. 5a and linear scale on Fig. 5b). The optimal values of $\gamma$ strongly depart from the default value of 60 days used in Crocus. The highest values of $\gamma$, in the 400 – 800 days range, were found at the Canadian Arctic sites of Bylot and Trail Valley Creek. Optimal values ranging around 100 – 200 days were found at Sodankylä, a site in northern Scandinavia, Umiujaq, a low Arctic site in Canada and at the high-elevation alpine site of Weissfluhjoch. Values close to the default 60 days values were optimal at Senator Beck and

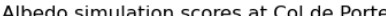

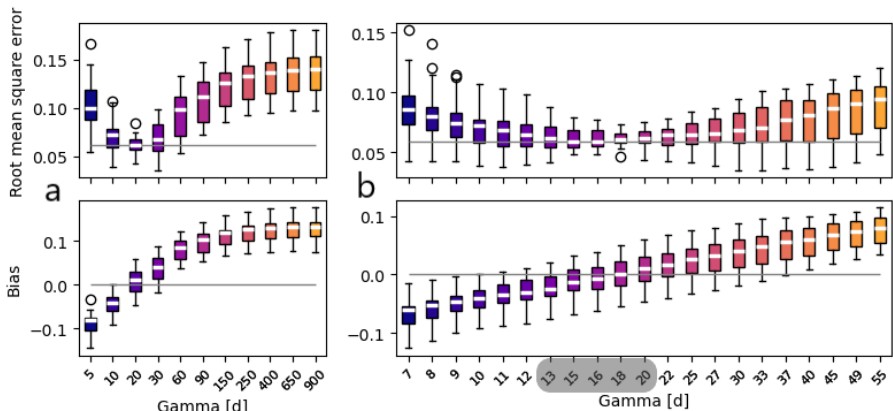

**Figure 4.** Box plot showing the distribution of the yearly scores at the Col de Porte site from 1994 to 2014, for the first round of evaluation (a) over the [5, 900] days range and (b) for the second round of evaluation zoomed in the [7, 55] days range. The median over the years for each $\gamma$ is shown in white. The box plots show the interquartile values, and outliers are plotted as circles. The grey rectangle along the $x$ axis shows the best $\gamma$ range derived from the evaluation process. Note that the vertical axes are not identical for plots (a) and (b).

Swamp Angel, the two high elevation Colorado sites. Finally, the lowest values of $\gamma$ appeared at Col de Porte, Sapporo, and Kühtai.

## 4.2   Climatology of light-absorbing particles deposition on snow

### 4.2.1   Global climatology

From the GMASI product, a probability of snow cover at each location for each day of the year was computed, by dividing the
number of years where there was snow cover at the location by the number of years in the dataset. Summing these probabilities over all days of the year yielded the average number of snow cover days per year at each location over the 1988-2016 period. Fig. 6 shows the spatial variability at a global scale with a strong latitudinal trend in the Northern Hemisphere. Mountainous areas (mostly the Himalayas, the European Alps, and the Canadian and US Rockies) appear very clearly as they are more often snow-covered than the lower-elevation surrounding regions. As expected, Antarctica and Greenland are characterized by
the presence of snow cover that extends all year long. The no-snow mask applied in the Southern Hemisphere in the GMASI product (see Sect. 3.1.2) can be clearly seen in South America (east of the Andes), in South Africa and in Australia.

Global maps of mean annual dust and BC deposition rates on snow are presented in Fig. 7. Some spatial patterns are visible: BC deposition is higher in South-East Asia and Europe (Fig. 7a) whereas low BC deposition rates are found in the Canadian Arctic and in Eastern Siberia. Mineral dust is mainly deposited in regions close to the main source regions such as the Middle
East, the Andes (close to the Altiplano and Patagonian deserts), and to a lesser extent the West of the United States (Fig. 7b). The lowest deposition rates of BC and dust on snow are found in Antarctica. Mineral dust deposition rates are overall higher

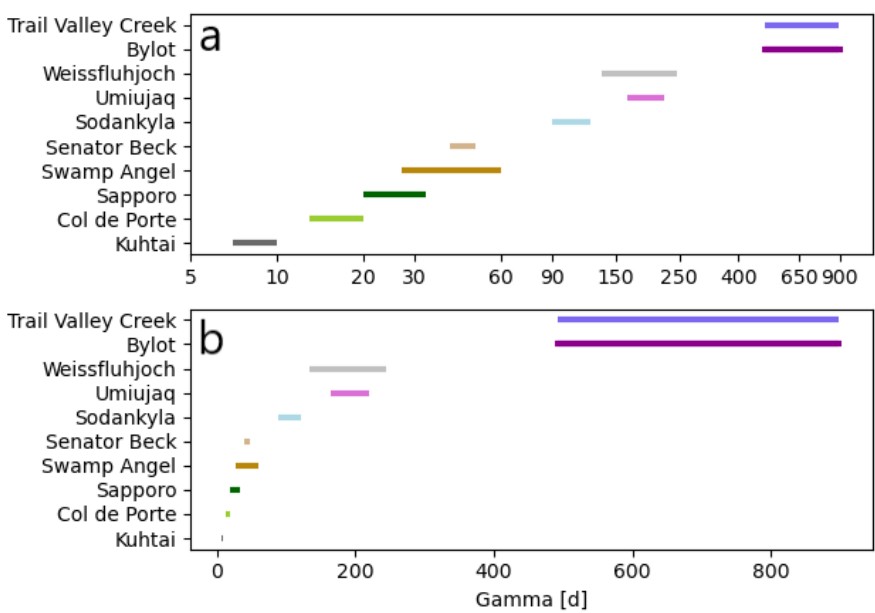

**Figure 5.** Optimal range of $\gamma$ at each site, in (a) logarithmic and (b) linear scale.

than BC deposition rates, but the impact of mineral dust on albedo is much lower than the impact of BC (about three orders of magnitude according to Eq. 6; Clarke et al. (2004)). For this reason, the total LAP deposition rates on snow (Fig. 7c) have similar orders of magnitude to BC deposition rates, with a spatial variability that reflects the variability found in the climatology

of both dust and BC deposition on snow.

Figure 8 quantifies the inter-annual variability of BC, dust and total LAP deposition on snow (Eq. 7). The Mediterranean basin, Turkey, the Caucasus, the southern Andes and the mountains of Western US present a large inter-annual variability of dust deposition on snow associated with the strong variability of dust deposition events in these regions (Skiles et al., 2012; Di Mauro et al., 2015; Dumont et al., 2020; Réveillet et al., 2022; Haugvaldstad et al., 2024). The inter-annual variability of dust

deposition on snow is generally larger than the inter-annual variability of BC deposition on snow, in agreement with Réveillet et al. (2022). Antarctica appears as the region with the strongest inter-annual variability of BC deposition on snow (compared to the climatology) but this variability remains the lowest around the planet when considered in absolute value (Fig. S4 in the Supplementary Material). As for the total deposition rates, the inter-annual variability of total LAP deposition on snow remains close to the values found for BC, with the exception of regions where the inter-annual variability of dust deposition on snow is

the largest (Karakum Desert, Aral Sea, Gobi Desert and south-western Andes).

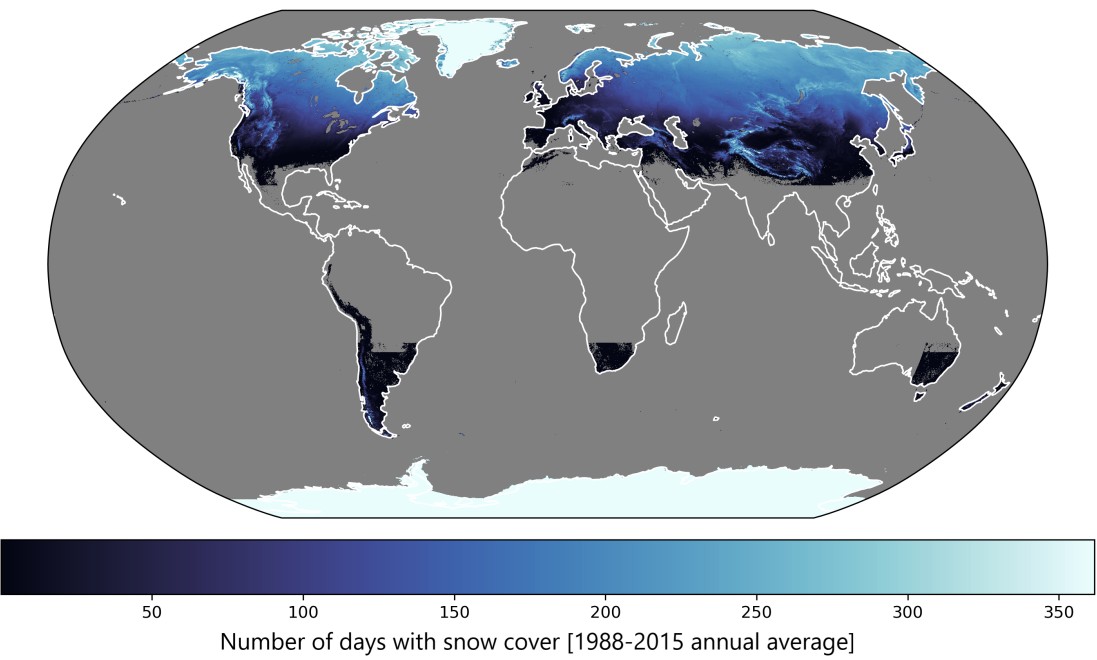

**Figure 6.** Global map of number of days with snow cover (1988-2015 annual average), derived from the GMASI dataset.

### 4.2.2 Climatology at the reference sites

The local gradients of total LAP deposition with elevation were computed around each mountainous site, as described in Sect. 3.2.2, and are shown on Fig. 9. Senator Beck and Swamp Angel are represented on the same figure (Fig. 9c) since they fell within the same grid cell. The comparison of the actual elevation at each site with the elevation of the corresponding grid cell in the GFDL dataset, as well as the value of the gradient and the regression coefficient of each linear regression are also shown in Fig. 9. These altitudinal gradients revealed contrasted evolutions of LAP deposition rates with elevation around the different sites. Indeed, the two sites located in the Rockies display an increase in LAP deposition as a function of elevation whereas a decrease with elevation is found for the three sites located in the European Alps with similar values of the altitudinal gradient.

The LAP climatologies were then extracted from Fig. 7c at the ten experimental sites considered in the study and were adjusted using the altitudinal gradients for the five mountainous sites. Fig. 10 presents the initial and adjusted values of LAP deposition rates for all sites. The sites displayed a wide range of LAP deposition rates, with over two orders of magnitude of difference between the least contaminated site (Bylot in Northern Canada) and the most contaminated (Sapporo in Japan). The Canadian Arctic sites presented the largest inter-annual variability of LAP deposition on snow (up to one order of magnitude at Umiujaq) and received significantly lower LAP deposition than the rest of the sites. The sites in the European Alps had the highest deposition rates after Sapporo while the two Colorado sites as well as Sodankylä received mid-range deposition rates. The largest altitudinal correction was performed at the Weissfluhjoch site due to the large difference between the actual

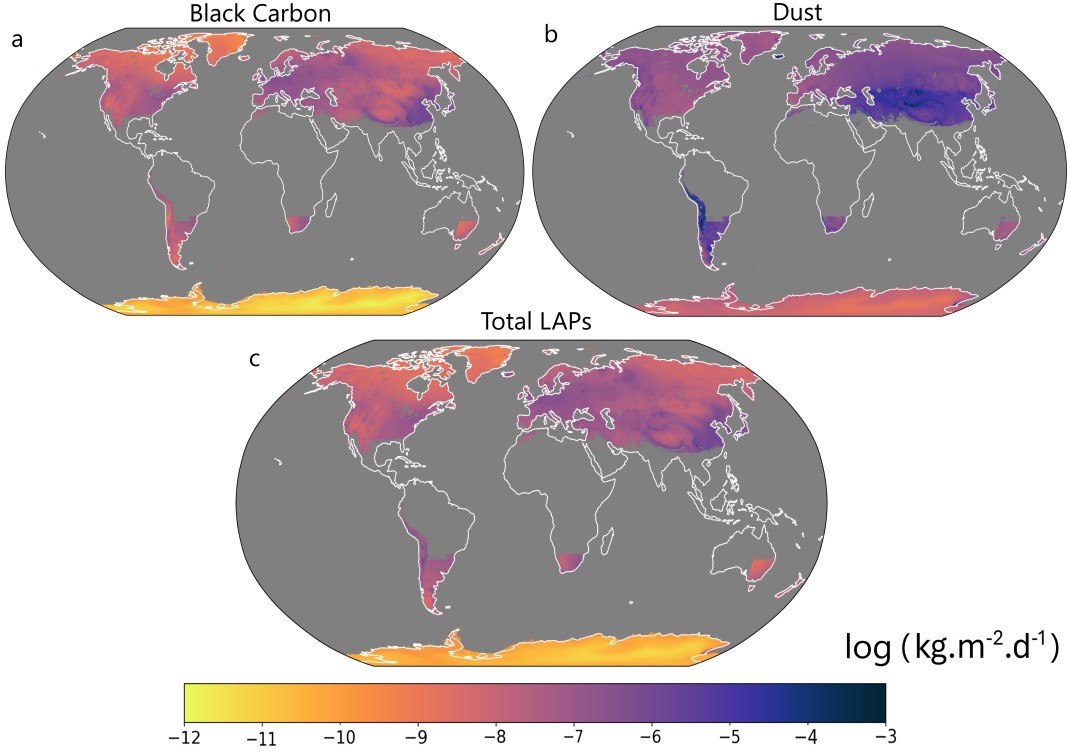

**Figure 7.** Global maps of annual mean (a) BC, (b) dust, and (c) total LAP (BC + dust in equivalent BC) deposition rates over snow. The logarithmic color scale highlights the orders of magnitude of difference between BC (a) and dust (b) deposition rates.

elevation of the site and the corresponding GFDL elevation. Only minor corrections were applied to the other mountain sites. At Swamp Angel and Senator Beck, the correction was positive whereas it was negative at the other sites, in agreement with the altitudinal gradients shown on Fig. 9.

### 4.3 Cross-analysis

The linear regression method presented in Section3.3 was applied to find a relationship between the range of statistically equivalent values of $\gamma$ and the climatological LAP deposition at each site. It is presented in Fig. 11, before and after the correction for elevation was applied to the mountainous sites. The regression with the altitudinal correction had the following characteristics:

$$log(\gamma) = -0.641 \log(D_{\mathrm{mean,LAP}}) - 2.99, R^2 = 0.642, \tag{10}$$

where $D_{\mathrm{mean,LAP}}$ is the climatological deposition rate of total LAPs over snow expressed in kg m$^{-2}$ d$^{-1}$ of equivalent BC. The regression without altitudinal correction was very similar with a slightly lower regression coefficient of $R^2 = 0.613$.

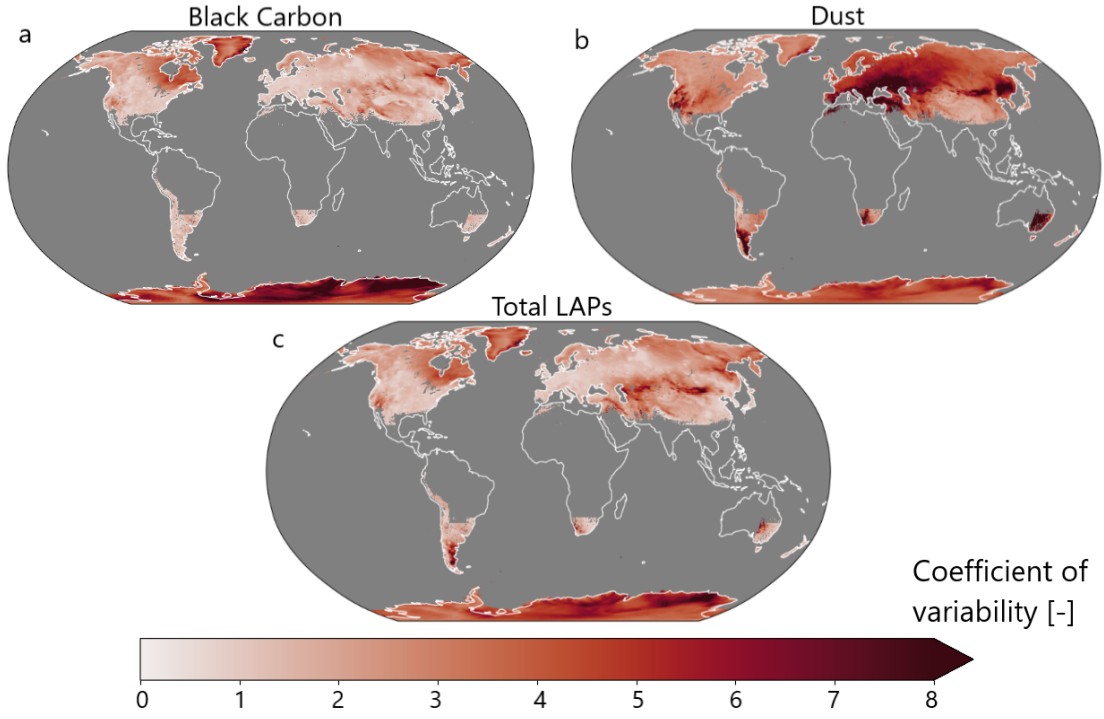

**Figure 8.** Global maps of the coefficient of variability of annual mean (a) BC, (b) dust, and (c) total LAP (BC + dust in equivalent BC) deposition rates over snow between a high and a low LAP year.

The two linear regressions are very similar (Fig. 11), but the correction for elevation provided a slight improvement in the regression coefficient, so that this regression is used as the reference regression in the rest of this study. Updates values of $\gamma$ (referred to as $\gamma_R$) were obtained at each site from Eq. 10.

The benefit of using $\gamma_R$ compared to the current default in Crocus (60 days) was then assessed in terms of simulated snow albedo and snow depth at the 10 experimental sites. Results are shown on Fig. 12. Improvements in snow albedo simulations (i.e., decrease in RMSE) are found at 7 sites (Fig. 12a). The average relative decrease in RMSE over the 10 sites was 10% for snow albedo. The albedo simulations were improved most at the three Canadian Arctic sites (decrease in RMSE larger than 25%), because the default value of 60 days was very small compared to the value of $\gamma_R$ at these sites (Fig. 11). At Col de Porte, Kühtai and Sapporo, the opposite effect was observed: the default value was too high compared to $\gamma_R$ and the ideal range of $\gamma$ at these sites. In Sodankylä, $\gamma_R$ fell very close to the ideal range of $\gamma$ at this site and provided a moderate improvement in albedo simulations (decrease in RMSE by 6 %). Albedo simulations at the two Colorado sites and at Weissfluhjoch were degraded when using $\gamma_R$, which is further away from their optimal ranges of $\gamma$ than the 60 days default (Fig. 11).

The snow depth (SD) simulations show different improvement tendencies compared to snow albedo (Fig. 12b) with an average relative decrease in RMSE over the 10 sites of 3% (10 % for albedo). SD simulations were improved at 6 out of the 10 experimental sites. Among these 6 sites, 5 of them showed simultaneous improvements in albedo and SD simulations. SD

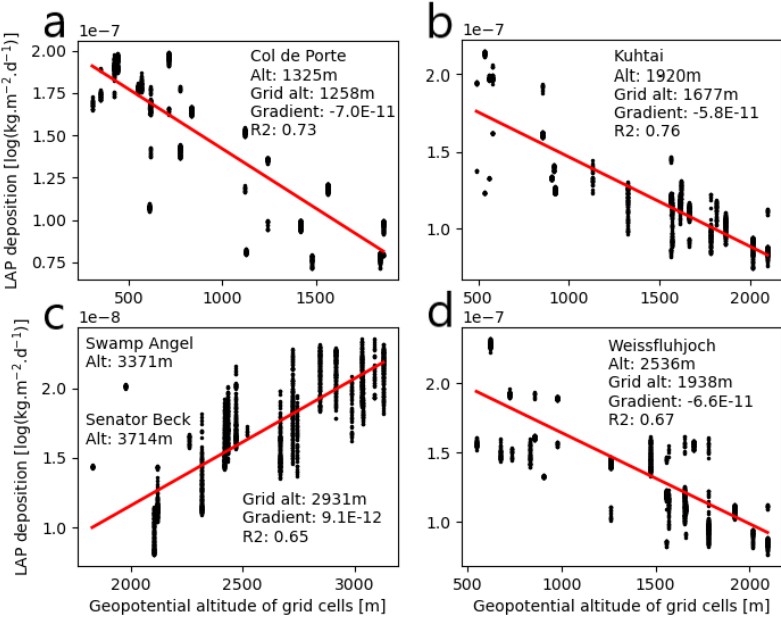

**Figure 9.** Local gradient of the total LAP climatological deposition rate with elevation and associated linear regression at (a) Col de Porte, (b) Kühtai, (c) Swamp Angel and Senator Beck, and (d) Weissfluhjoch. Note that the axes are not identical between the plots. The gradients have the units kg m$^{-2}$ d$^{-1}$.

simulations at Col de Porte were degraded when using $\gamma_R$ (increase in RMSE by 10%) despite strong improvements in albedo simulations (decrease in RMSE by 13%). A similar behavior was observed for the SD simulations at Sodankylä. Using $\gamma_R$
degraded SD simulations at Weissfluhjoch (increase in RMSE by 12%) in agreement with the degradation found for snow albedo (increase in RMSE by 13%). The largest RMSE for SD simulations (using both configurations of the model) was found at Senator Beck. This site is exposed to wind-induced snow redistribution which cannot be reproduced by the model in point-scale mode. The same issue affected a large number of models used in the ESM SnowMIP exercise (Menard et al., 2021).

Fig. 13a shows the global map of $\gamma_{\mathrm{mean}}$ derived from Eq 10. The horizontal resolution of this global map is the same as the
global climatology of LAP deposition on snow (0.04° by 0.04°). Threshold values for $\gamma_{\mathrm{mean}}$ were set at 5 and 900 days. The lowest values of $\gamma_{\mathrm{mean}}$ (in the range 5 to 50 days) should be used in the regions that receive the largest deposition of LAP, notably South-East Asia and Europe (Fig. 7c). The highest values of $\gamma_{\mathrm{mean}}$ (above 500 days) should be used in regions which receive low LAP depositions, namely Antarctica, Greenland and the high Canadian Arctic. The largest uncertainties for $\gamma$ as shown on Fig. 13b are found in regions where the inter-annual variability of LAP deposition on snow is the largest (Fig. 8c).
This includes the east coast of North America and central Siberia that are close to regions of BC emissions. The Karakum Desert, the Aral Sea and the Gobi Desert are also regions of large variability in $\gamma$ due to the strong inter-annual variability of dust deposition in these regions (Fig. 8b). Finally, Antarctica shows no inter-annual variability of $\gamma$ since the values of $\gamma$ for

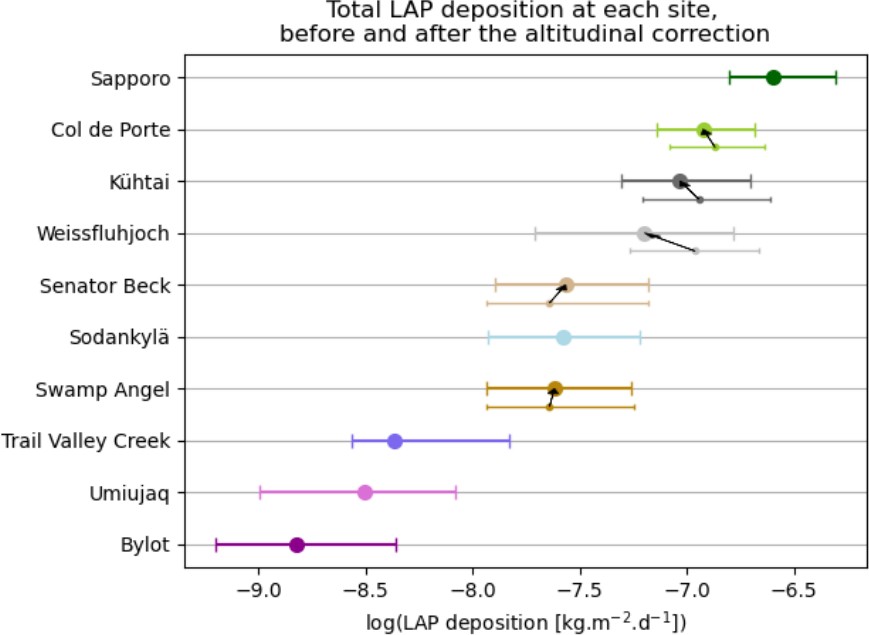

**Figure 10.** Graph representing climatological LAP deposition rates at all sites, before and after the correction of elevation effects. Arrows indicate these corrections, at mountainous sites. The error bars represent the inter-annual variability of LAP deposition at each site.

a low and high LAP in this region are equal to 900 days (the maximal value allowed for $\gamma$ in Crocus) due to very low LAP deposition on snow (Fig. 7c).

## 5 Discussion

Multiyear simulations with the Crocus snowpack scheme have been carried out to determine the ranges of values of the snow darkening coefficient $\gamma$ giving the best snow albedo simulations at ten sites covering different climates (Fig. 2 and Table 2). A large variability of the optimal range of values for $\gamma$ was found across the ten sites (Fig. 5). These results confirm that a single value of $\gamma$ cannot be used for all climates around the world, and this parameter needs to be tuned as proposed by Brun et al. (2011). In particular, the values in the 500- to 900-day range found in this study for the two sites located in the Canadian Arctic (Trail Valley Creek and Bylot Island) are consistent with the values used for polar snow by Brun et al. (2011) and Woolley et al. (2024). On the other hand, Fig. 5 reports optimal $\gamma$ values in the range 13-20 days at the Col de Porte experimental site. These values differ from the default value of 60 days that was optimized by Brun et al. (1992) for this site. However, Brun et al. (1992) did not use albedo measurements to tune the value of $\gamma$. Instead, they relied on measurements of surface temperature to show that $\gamma$=60 days provided good simulations of the snow surface energy balance. Lafaysse et al. (2017) have shown that the energy balance simulated by Crocus is associated with additional sources of uncertainties, in particular due to the formulation

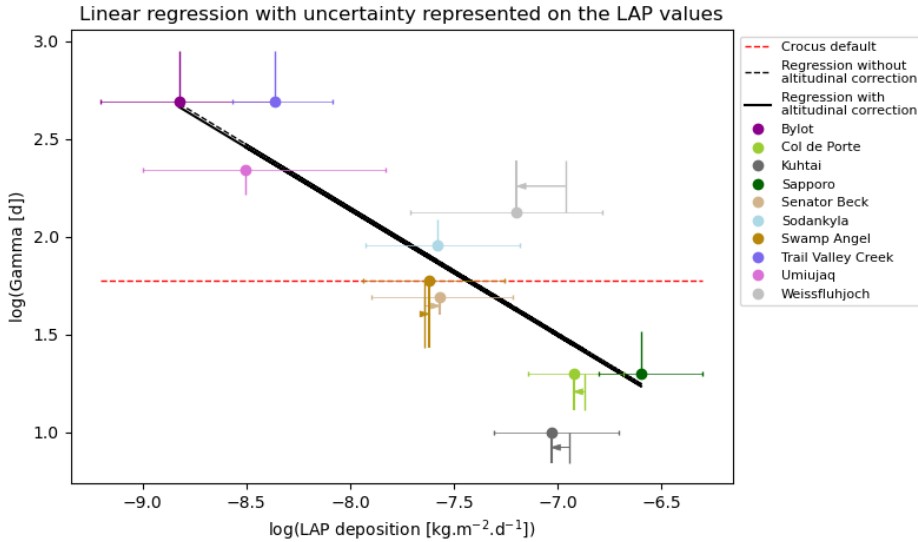

**Figure 11.** Graph representing the optimal ranges of $\gamma$ as a function of total LAP deposition rates in equivalent BC, in logarithmic scales, and the corresponding linear regressions with and without correction for elevation at the mountain sites (the correction is represented by the arrows). The points which have been represented with circles in each range are the ones closest to the final regression (Eq. 10), which are used to compute the regression coefficient (see method in Appendix C). The error bar along the x-axis represents the inter-annual variability of LAP deposition at each site.

of turbulent fluxes (Martin and Lejeune, 1998). The direct optimization of $\gamma$ on snow albedo proposed in this study reduces the impact of these additional sources of uncertainties on the estimation of the optimal values of this parameter.

The optimal values of $\gamma$ shown in Fig. 5 have then been combined with climatological values of LAP deposition on snow at the ten experimental sites to determine a regression between these two variables (Fig. 11). This logarithmic linear regression confirms the dependency of $\gamma$ on the LAP deposition rate which influences snow albedo in the visible range. 64% of the variation in $\gamma$ is explained by the regression model. The largest discrepancies between the regression and the optimal values of $\gamma$ are found for the Weissfluhjoch and Kühtai sites, located in the European Alps. These two sites are geographically close to each other (approximately 150 km apart) with optimal values of $\gamma$ in the range 134-245 days at the Weissfluhjoch site and in

the range 7-10 days at Kühtai. According to the GFDL climate model, these two sites present similar climatological values of LAP deposition on snow. The correction for elevation proposed in this study only slightly differenciated these climatological values. The discrepancy between the two sites may result from the inability of the global climatology of LAP deposition on snow to capture fine-scale features of LAP deposition in mountainous terrain and the presence of local LAP sources. The Weissfluhjoch site is located well above an alpine valley whereas the Kühtai snow site is located in a valley. The GFDL climate

model at 50-km grid spacing (Zhao et al., 2018) cannot accurately reproduce orographic precipitations, which are essential for wet deposition of LAPs, nor channelling of dust flow through valleys (Baladima et al., 2022).

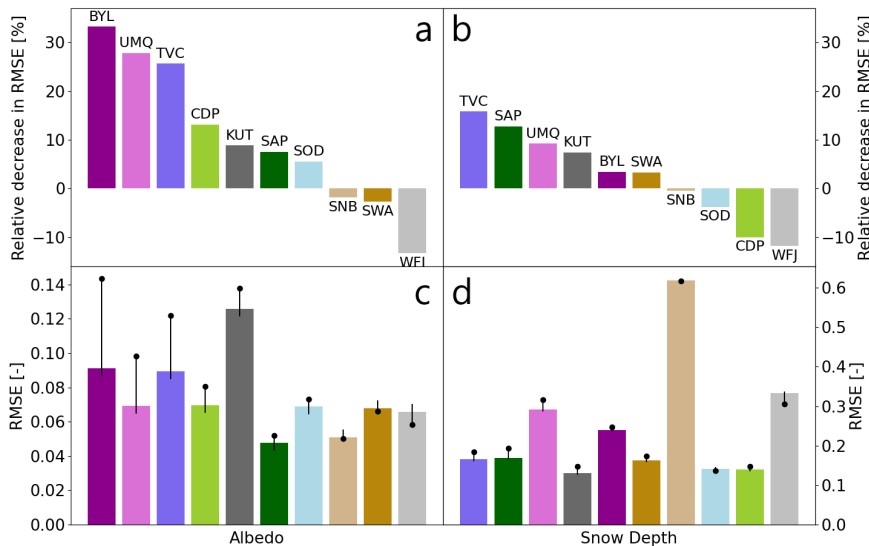

**Figure 12.** For each site, relative improvement of simulation skills for (a) albedo and (b) snow depth, from this study compared to the default 60 days value, and absolute improvements for (c) albedo and (d) snow depth, where the black dots are the default 60 days value and the bars are the results from this study.

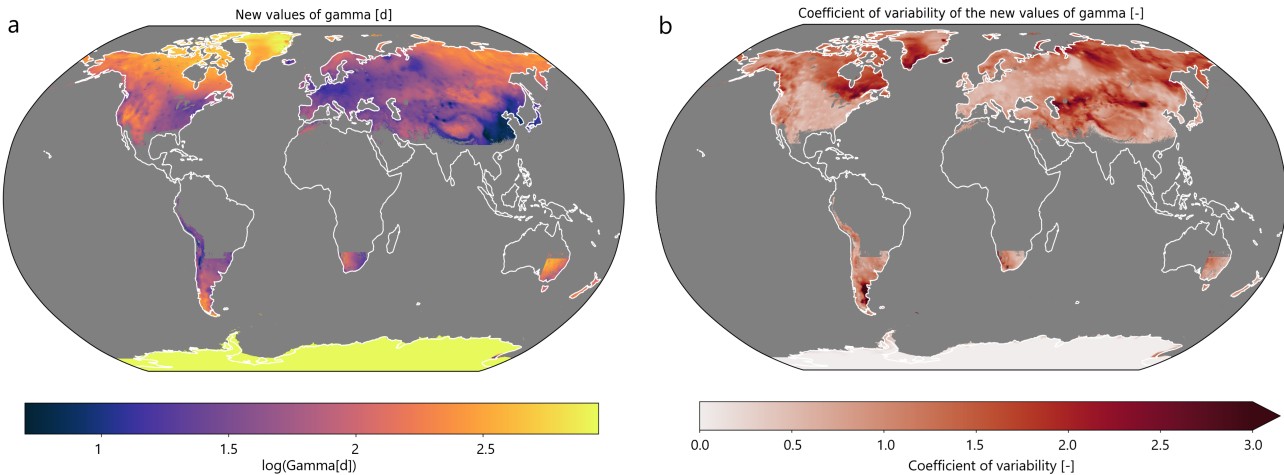

**Figure 13.** Global maps showing (a) the optimal value of $\gamma$ derived from Eq. 10 and (b) the coefficient of variability of $\gamma$ associated with the inter-annual variability of LAP deposition on snow.

The climatology of LAP deposition on snow would therefore benefit from inputs of chemistry transport models at 10-km resolution for global applications (e.g. Bessagnet et al., 2017) and at even higher resolutions for regional applications (e.g.

Baladima et al., 2022). Such products would allow to refine the altitudinal adjustments applied to the local climatology to
correct for the difference of elevation between the climatology and the actual elevation of the sites. The results obtained with
the current climatology suggest that the local altitudinal gradients of LAP climatological deposition rate are characteristic
of each mountain range (Fig. 9). The climatological LAP deposition rate decreases with elevation in the Europeans Alps
(around Weissfluhjoch, Col de Porte, and Kuthai) whereas it increases with elevation in the Colorado Rocky Mountains (USA,
around Senator Beck and Swamp Angel). The default snow albedo parameterization in Crocus is consistent with the altitudinal
gradient found in the European Alps since it uses a pressure term (Eq. 2) to increase the value of $\gamma$ with elevation, indirectly
representing a decease in LAP deposition with elevation. However, this term is not suitable for the Colorado Rocky Mountains
nor potentially for other mountain ranges. This result justifies why the pressure term was not considered in this study and
highlights the need to carefully revise this term when Crocus is applied in different mountain ranges across the world.

The LAP deposition outputs in the GFDL dataset only contain the deposition rates for BC and dust as percentiles for each
day of the year, based on the 1979-2015 period. This characteristic of the dataset led to two limitations of this study. First,
the inter-annual variability of LAP deposition on snow was calculated for a low LAP year and a high LAP year using the
percentiles 25 and 75 for dust and BC deposition for each day of the year. Such a method can only provide an estimation
of the inter-annual variability of LAP deposition. A more accurate computation would have required to use the deposition
rates of BC and dust for each day of the year and to combine them with the corresponding snow cover information. Mean
daily deposition of LAP on snow could have then been computed for each year of the period of interest and the inter-annual
variability could have been estimated precisely. Second, the regression between $\gamma$ and the LAP climatological deposition rates
relied on a total of 10 points corresponding to the ten experimental sites considered in this study (Fig. 11). The size of the
sample limits the robustness of the regression. Having average LAP deposition rates on snow for each year and each site would
have allowed a comparison with the corresponding optimal values of $\gamma$ for each year and have increased the size of the sample
considered in the regression ten-fold. To overcome these limitations, the MERRA2 atmospheric reanalysis (Gelaro et al., 2017)
could be considered as an alternative to the GFDL dataset since MERRA2 provides global daily estimates of wet and dry LAP
depositions rates at 50-km resolution since 1980. Adding new sites in the analysis is another option to increase the size of this
sample in the regression, but it requires reference sites with quality-controlled meteorology and dedicated snow observations
such as those available in the ESM-SnowMIP dataset (Ménard et al., 2019).

The values of $\gamma$ determined with the logarithmic linear regression improved snow albedo simulations with Crocus at seven
out of the ten sites (Fig. 12a and c), with the largest improvements found for the Arctic sites (decrease in RMSE larger
than 25%). These results confirm the robustness of the approach at the sites where the regression was developed. Further
evaluations using independent sites are required to confirm the transferability of the approach. Simulations of snow depth with
Crocus including the LAP-dependent $\gamma$ were also slightly improved compared to the default version of Crocus (Fig. 12b
and d; average decrease in RMSE of 3% at the ten sites). However, certain sites such as Col de Porte and Sodankylä showed
degradations in the simulations of snow depth despite improvements in the simulation of snow albedo. This behavior may
be explained by the fact that the quality of snow depth simulations by the model is influenced by other sources of errors in
the different model parameterizations (Lafaysse et al., 2017; Günther et al., 2019). Ensemble snowpack simulations (Lafaysse

et al., 2017) could be used to quantify the uncertainties associated with these different parameterizations and identify the role of the adjusted values of $\gamma$. Despite the improvements brought in this study, the default snow albedo in Crocus still suffers from limitations associated with the fixed ratio used to compute the broadband albedo from the values in the three spectral bands (Eq.1) and the absence of effect of the solar zenith angle on snow albedo. Gardner and Sharp (2010) have shown that changes in solar zenith angle and clouds' optical thickness can lead to changes on the order of 0.05 in the snow albedo values (Fig. 9 in their study). Including these effects will lead to further improvements in large scale snow albedo simulations with Crocus.

The methodology developed in this study relies on climatological values of LAP deposition on snow, so that it cannot represent the seasonal variation of LAP deposition on snow and the impact of individual LAP deposition events that can strongly influence the evolution of snow albedo  (e.g. Di Mauro et al., 2015; Dumont et al., 2020). Their representation can only be achieved by coupling a snowpack model with a more numerically expensive snow radiative transfer model and using LAP deposition fluxes as additional forcing (Flanner et al., 2007; Tuzet et al., 2017; Réveillet et al., 2022). Instead, the methodology proposed in this study aims at capturing the impact of the large spatial variability of LAP deposition on the simulation of snow albedo. It has been applied to the default albedo parameterization of the Crocus snow scheme and presents a strong potential to improve continental-scale snow simulations (Brun et al., 2013; Mortimer et al., 2020), which serve as a basis for climate trend-analysis (Mudryk et al., 2024). The uncertainty in $\gamma$ resulting from the inter-annual variability of LAP deposition on snow varies spatially (Fig. 13b) and can be considered in the context of ensemble snowpack simulations. The methodology developed in this study can be applied to optimize parameters that indirectly represent the impact of LAPs on snow albedo in the visible range in other snowpack models explicitly including the effect of optical grain size on albedo (Dickinson et al., 1993; Best et al., 2011; Decharme et al., 2016). The time constants in more simple time-dependent snow albedo parameterizations (e.g., Verseghy, 1991; Douville et al., 1995; Pedersen and Winther, 2005) could also be optimized using the global climatology proposed in this study. However, this optimization would require a careful analysis to make the distinction between the combined effects of several snow aging processes contributing to the decrease of snow albedo with time, such as LAP depositions and the increase in optical grain size due to snow metamorphism.

This study has established a relationship between the snow darkening coefficient ($\gamma$) and the climatological values of LAP deposition on snow. Several factors that are influencing the spatial distribution of $\gamma$ are not included. Firstly, only deposition of dust and BC were considered when computing the climatology of mean LAP deposition over snow. The darkening effect of brown carbon (BrC) resulting from biomass burning and biofuel sources (Beres et al., 2020; Brown et al., 2022) is not included since BrC deposition rates are not available in the GFDL dataset. Brown et al. (2022) found that the ratio of globally averaged snow darkening effect from BrC to the one from BC ranges from 37% to 98%. This suggests that including BrC could help refine the global map of $\gamma$ shown on Fig.  13. In addition, the effects of forest litter and debris on snow albedo in the visible range and on the resulting $\gamma$ values (Melloh et al., 2001) were not considered in this study. Only sites located in open terrain were selected to determine the optimal ranges of $\gamma$. Therefore, the values of $\gamma$ shown on Fig. 13 do not include the effect of the forest presence. The approach of Hardy et al. (2000) could be considered to indirectly represent the effects of forest litter in the default snow albedo scheme used in Crocus.

## 6 Conclusions

The goal of this study was to develop a methodology of intermediate complexity to improve large scale simulations of snow albedo in snowpack schemes by taking into account the spatial variability of LAP deposition (BC and dust). Toward this goal, a global climatology of LAP deposition over snow has been built by combining a climatology of daily LAP deposition from a global climate model with a global remotely-sensed snow cover dataset to retain only deposition over snow, providing an estimation of the average BC and dust deposition rates over snow for the period 1981-2015 and an estimation of the associated inter-annual variability. The two types of LAPs were then expressed in terms of equivalent black carbon and summed to obtain the climatology of total LAP deposition. This climatology and the associated inter-annual variability are available for scientific use (Gaillard et al., 2024a).

The default albedo parameterization in the detailed snowpack scheme Crocus has then been improved using this climatology by optimizing a parameter that controls the snow albedo evolution in the visible range (known as the snow darkening coefficient, $\gamma$). This coefficient implicitly represents the darkening of the snow with time due to LAP deposition with low (high) values indicative of darkened (clean) snow. Multi-year snowpack simulations were carried out with Crocus at ten reference sites covering a large variety of climates. The range of optimal values of $\gamma$ yielding the most skilled albedo simulations was found at each site, accounting for uncertainties in the observation of snow albedo. This analysis revealed a wide variety and a good dispersion of the ranges of $\gamma$ at the ten chosen sites: ideal $\gamma$ values went from 7 days at Kühtai, in the Austrian Alps, to 898 days at Bylot and Trail Valley Creek, in the Canadian Arctic.

A logarithmic linear regression was then applied between the optimal ranges of $\gamma$ and the LAP deposition rates at the ten sites extracted from the climatology. These LAP deposition rates ranged over two orders of magnitude between the Canadian Arctic sites and Sapporo in Japan. The logarithmic linear relationship was finally combined with the global climatology to obtain an LAP-informed and spatially variable $\gamma$ parameter for the Crocus albedo parameterization. The revised parameter improved, on average, snow albedo simulations by 10% with the largest improvements found in the Arctic (more than 25%). The uncertainty in $\gamma$ resulting from the inter-annual variability of LAP deposition on snow was computed. The global dataset with the optimal values of $\gamma$ and the associated uncertainty are available for the users of Crocus (Gaillard et al., 2024b). This approach takes into account the climatological spatial variability of LAP deposition on snow but cannot represent the seasonal variation of LAP deposition which can result from single deposition events, nor the inter-annual variability of LAP deposition.

Future work will test the impact of the spatially optimized values of $\gamma$ in snowpack simulations with Crocus over large domains such as Canada. The methodology detailed in this paper can be applied to optimize parameters controlling the albedo evolution in the visible range in other snowpack schemes (e.g., Dickinson et al., 1993; Best et al., 2011; Decharme et al., 2016) without requiring coupling with a more expensive snow radiative transfer model.

*Code and data availability.* The global climatology of LAP deposition on snow is available at https://doi.org/10.5281/zenodo.11554783 (Gaillard et al., 2024a). The LAP-informed $\gamma$ values are available at https://doi.org/10.5281/zenodo.11554926 (Gaillard et al., 2024b). The code of Crocus within the land surface scheme SVS2 used in this study is available at https://github.com/VVionnet/MESH_SVS/tree/master.

**Table A1.** Physical options of the multiphyscis version of Crocus used in this study. The names of the options refer to the Crocus namelist (Lafaysse et al., 2017)

| Physical process | Name of the option | Reference |
|---|---|---|
| Snowfall density | V21 | Vionnet et al. (2012) |
| Metamorphism | B21 | Baron (2023) |
| Turbulent fluxes | RI2 | Lafaysse et al. (2017) |
| Thermal conductivity | Y81 | Yen (1981) |
| Liquid water holding | B92 | Brun et al. (1992) |
| Compaction | B92 | Brun et al. (1992) |
| Snowdrift | VI13 | Vionnet et al. (2013) |

Integration of these developments in the version of Crocus available in the SURFEX modelling platform will be done after the publication of this study.

## Appendix A: Parameterizations used in Crocus

The multiphysics version of Crocus (Lafaysse et al., 2017) offers different options to simulate physical processes that drive the evolution of the snowpack. Table A1 details the options used in Crocus for all the simulations presented in this study.

## Appendix B: Method to select statistically-equivalent intervals of $\gamma$

For a given site, the score estimator for each value of $\gamma$ is:

$$\widehat{RMSE_{site,\gamma}} = \mathcal{M}_{years}(RMSE_{site,year,\gamma}), \tag{B1}$$

where $\mathcal{M}_{years}$ is the median over all the years available at this site and $RMSE_{site,year,\gamma}$ is the yearly RMSE score as expressed in Eq. 4. This score is uncertain because of the observational error. It is modelled as proposed in Lafaysse et al. (2017) using a normal distribution as a first-order approximation:

$$\mathbb{N}_{site,\gamma} = \mathcal{N}(\widehat{RMSE_{site,\gamma}}, \sigma_{RMSE}), \tag{B2}$$

where the variance $\sigma_{RMSE}$ is the estimated observational uncertainty corresponding to the RMSE score. The a priori value for $\sigma_{RMSE}$ suggested in Lafaysse et al. (2017) is 0.069. This value was estimated at the Col de Porte site, whose incoming radiation receptors are equipped with automatic wipers to avoid the inexactitudes from snowfall accumulation on the sensors. Most of the other sites in this study are not similarly equipped, though some have frequent manual wiping of the sensors. Therefore, this value of $\sigma_{RMSE}$ is potentially underestimated for most of the sites. In this study, it was nonetheless retained as a first order estimate of the observational uncertainty corresponding to the RMSE score. Numerically, 1001 values of $\mathbb{N}_{site,\gamma}$

were randomly generated to represent the distribution.

The statistically-equivalent range of $\gamma$ was finally computed by considering $\tilde{\gamma}$ the value of $\gamma$ which gave the best score estimator $\widehat{RMSE}_{site,\gamma}$. An independent samples t-test was applied between $\mathbb{N}_{site,\tilde{\gamma}}$ and each of the other distributions $\mathbb{N}_{site,\gamma}$, to assess if the difference between them was statistically significant or not. For this test, a confidence interval of 90% was used, as suggested in Lafaysse et al. (2017). All values of $\gamma$ which were not declared statistically different from $\tilde{\gamma}$ by the t-test were added to the range of best $\gamma$ values for the considered site. A range of statistically equivalent ideal $\gamma$ values was obtained for

each site and considered in the rest of the study.

## Appendix C: Method for the linear regression

This method slightly differs from classic linear regression methods that do not consider a uniform range of statically equivalent values for the target variable. The cost function used was inspired from the mean squared error (MSE), defined as follows for an independent variable $x$ and a dependent variable $y$:

$$MSE = \frac{1}{N} \sum_{i=1}^{N} (ax_i + b - y_i)^2, \tag{C1}$$

where the scatter plot from which to derive the regression is $(x_i, y_i)_{i \in [1,N]}$, and $a$ and $b$ are the regression coefficients such that $y = ax + b$. This MSE was therefore the sum of squared distances between the predicted value of $y$ and the actual value of $y$. To account for the ranges of $y$ (which in our study represents $\gamma$), the actual value of $y$ was replaced with the range of values $Y$:

$$MSE = \frac{1}{N} \sum_{i=1}^{N} \Delta_i^2, \tag{C2}$$

$$\text{with} \quad \Delta_i = \begin{cases} 0 & \text{if } y_{i,min} \leq ax_i + b \leq y_{i,max} \\ ax_i + b - y_{i,min} & \text{if } y_{i,min} \geq ax_i + b \\ ax_i + b - y_{i,max} & \text{if } y_{i,max} \leq ax_i + b \end{cases} \tag{C3}$$

where $y_{i,min}$ and $y_{i,max}$ are the minimal and maximal values within the range $Y_i$. A gradient descent method was used to optimize $a$ and $b$ following this cost function, using a learning rate of 0.01, 1000 iterations, and initial $a$ and $b$ values derived from an automatic standard linear regression for a random value within each range of $\gamma$.

Once the optimal a and b were found, the points closest to the regression within each range were found. These are the points which were taken into account during the final iteration of the gradient descent. If the regression had been carried out over these points instead of the ranges $Y$, the resulting $a$ and $b$ would have been the same. For this reason, these points were used to compute the regression coefficient $R^2$:

$$R^2 = 1 - \frac{\sum_{i=1}^{N} (ax_i + b - y_i)^2}{\sum_{i=1}^{N} (ax_i + b - \overline{y_i})^2} \tag{C4}$$

with    $y_i = \begin{cases} ax_i + b & \text{if } y_{i,min} \leq ax_i + b \leq y_{i,max} \\ ax_i + b - y_{i,min} & \text{if } y_{i,min} \geq ax_i + b \\ ax_i + b - y_{i,max} & \text{if } y_{i,max} \leq ax_i + b \end{cases}$      (C5)

where $\overline{y_i}$ is the average of $(y_i)_{i \in [1,N]}$. This coefficient was useful to evaluate the quality of the linear regression.

*Author contributions.* MG, VV, ML and MD were responsible for designing the methodology of the study. MG carried out the numerical simulations and the analysis. VV supervised the project. MD provided expertise on LAP deposition. ML advised on the statistical method to account for observation uncertainties in the optimal $\gamma$ values, and provided the meteorological forcing file for the Sodankylä site. PG

provided and advised on the GFDL dataset. MG and VV drafted the manuscript and all authors participated in reviewing and editing the paper.

*Competing interests.* Some authors are members of the editorial board of The Cryosphere.

*Acknowledgements.* The authors would like to thank Antoine Corcket, Georgina Woolley, and David Günther, who have provided some of the observation and meteorological forcing data for the reference sites of this study. Many thanks also to Gonzalo Leonardini for his help

on formatting the observation data at some of the reference meteorological sites. Marie Dumont has received funding from the European Research Council (ERC) under the European Union's Horizon 2020 research and innovation program (IVORI; grant no. 949516).

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
