# Peer review of "Improving large-scale snow albedo modeling using a climatology of light-absorbing particle deposition"

_EGUsphere, 2024_

## Author Comment (AC2)

**Answer to Reviewer 2**

We thank Reviewer 2 for their comments. We provide here our responses to those comments and describe how we will address them in the revised manuscript. The original reviewer comments are in normal black font while our answers appear in blue font.

This research project aims to represent the effect of Light-Absorbing Particles (LAP) on snow albedo within the Crocus model as snow aging parameters, and to establish a global climatology of LAP deposition on snow, with the intention of applying these results to land surface models. The effort to achieve a global climatology of this phenomenon, by accounting for the altitude dependence of snow aging and making it regionally specific, is commendable. However, there has been insufficient discussion regarding the methodology for estimating the parameters related to snow aging and LAP climatology when scaling this approach globally. Consequently, it cannot be stated that the reliability of the climatology related to LAP deposition, which is the final outcome, is fully assured. In particular, the following points must be thoroughly discussed:

(major comments)

1. As the author demonstrates, the parameters associated with snow aging show significant differences between the accumulation and melting periods (Fig. 3). However, this study only considered the regional dependency of snow aging. It would be more beneficial for the authors to evaluate snow aging separately for the accumulation and melting periods and then discuss the relationship between snow aging and LAP deposition on the snow. In other words, the climatology of LAP deposition will be influenced not only by altitude but also by the timing of the accumulation and melting periods.

As mentioned by Reviewer 2, Figure 3 in the submitted manuscript shows large changes in the temporal evolution of the simulated albedo with contrasted behavior during the snow accumulation and melting periods. This evolution is mainly explained by four processes simulated by the model:

1. The albedo decrease due to the increase in snow optical grain diameter resulting from dry snow metamorphism during periods without precipitation and no melting. This evolution of the snow optical grain diameter is handled by the metamorphism scheme for dry snow in Crocus (Carmagnola et al., 2014).
2. The albedo decrease due to the increase in snow optical grain diameter resulting from wet snow metamorphism during melting periods. The evolution of the snow optical grain diameter is handled by the metamorphism scheme for wet snow in Crocus (Carmagnola et al., 2014).
3. The albedo decrease in the visible range due to LAP deposition. This effect is empirically represented by the value of gamma in Crocus in the albedo parameterization.
4. The albedo increase due to snowfall which is handled by the snowfall module in Crocus.

The three first processes can be considered as snow aging processes (snow metamorphism and LAP deposition) that are responsible for the decrease of snow albedo with time. However, gamma only directly impacts the third aging process. Therefore, we believe that the term "snow aging coefficient" in the submitted paper was creating confusion for the reader. Reviewer 1 made a similar comment. For this reason, gamma will be renamed the "snow darkening coefficient" in the revised manuscript

since this coefficient is only used to empirically represent the impact of LAP deposition on the snow albedo in the visible range. We believe this new name will help the reader to better understand the role of gamma in the evolution of snow albedo in Crocus. More explanations about the snow albedo parameterization in Crocus will be added to the revised manuscript (Section 2):

*The evolution of the optical grain size in Crocus is computed using metamorphism laws as described in Brun et al. (1992) and Carmagnola et al. (2014). For a given layer, for dry snow, the temporal evolution of the optical grain size is a function of the vertical temperature gradient. whereas, for wet snow, the increase in optical grain size with time depends on the snow liquid water content. New snow in Crocus is characterized by an optical diameter of 1e-4 m (surface specific area of 65 m2/kg) and results in an increase in snow albedo.*

For the reasons mentioned above, we believe that the snow darkening coefficient should not be too different during the snow accumulation and melting periods since it is not associated with the physical processes that drive the large differences of snow albedo evolution during the accumulation and melting seasons. For example, the impact of the presence of liquid water on the snow optical grain size is handled by the metamorphism scheme in Crocus. Instead, gamma can be seen as an average quantity of LAP in the snowpack. The work proposed in this study allows spatially varying values of gamma based on the climatology of LAP deposition on the snowpack. However, it cannot present the interannual and seasonal variability of LAP deposition on the snowpack. It will be mentioned in the discussion and conclusion of the revised paper:

*The methodology developed in this study relies on climatological values of LAP deposition on snow, so that it cannot represent the **seasonal variation of LAP deposition on snow** and the impact of individual LAP deposition events that can strongly influence the evolution of snow albedo (Di Mauro et al., 2015; Dumont et al., 2020).* (Discussion of the revised paper)

*This approach takes into account the spatial variability of LAP deposition on snow but cannot represent the seasonal variation of LAP deposition that can result from single deposition events as well as the interannual variability of LAP deposition.* (Conclusion of the revised paper)

2. (L173) If the optical thickness of the snow cover is insufficient, the snow albedo will be influenced by the albedo (reflectance) of the ground surface beneath the snow cover. Consequently, γ (gamma) will also reflect the influence of the ground surface. In forested areas, the effect of vegetation must be considered as well. Therefore, γ is not only affected by snow aging but may also be strongly influenced by local factors. It is necessary to provide a comprehensive explanation of this point when determining the relationship between D (climatological deposition rate of LAPs) and γ.

As mentioned in our answer to the first point raised by Reviewer 2, the snow darkening coefficient gamma only represents the effect of LAP deposition on snow albedo in the visible band in Crocus. To avoid the effect of ground contamination on the measurement of snow albedo, a threshold on snow depth has been applied to remove periods when the snow cover was not thick enough. A threshold value of 20 cm was selected for the sites below 60 N and 10 cm was used for the sites above 60 N (L 175-177 of the submitted manuscript). Snow albedo calculations using the online

version of the snow radiative transfer model SNICAR v3 (Flanner et al., 2021) ([SNICAR-Online (umich.edu))](SNICAR-Online)) have been used to quantify the impact of the ground surface beneath the snow cover on broadband snow albedo for snow depth values corresponding to the threshold values selected in our study. Two snowpack have been considered: (i) a snowpack made of fresh snow with high SSA and low density values (typical of the accumulation season) and (ii) a snowpack made of melt forms with low SSA and high density values (typical of the ablation season). The SSA and density values for each type of snow were taken from Domine et al. (2012). The results are presented in Table 1 below. For a 10-cm (20-cm) thick snowpack made of fresh snow, the ground contamination reduces the snow albedo by 0.8 % (0.2 %). For a 10-cm (20-cm) thick snowpack made of melt forms, the ground contamination reduces the snow albedo by 1.9 % (0.6 %).

Table 1: Snow albedo computed by SNICAR for 2 different snowpack of various thickness.

| | Snow albedo values for different snowpack thickness | | |
|---|---|---|---|
| Type of snowpack and ground | Thickness = 0.1 m | Thickness = 0.2 m | Thickness = 100 m (optically infinite) |
| SSA: 65 m2/kg Density: 150 kg/m3 Ground albedo: 0.25 No LAP | 0.841090 | 0.846047 | 0.848279 |
| SSA: 10 m2/kg Density: 300 kg/m3 Ground albedo 0.25 No LAP | 0.772633 | 0.782954 | 0.787879 |

Based on this analysis, we can consider that the threshold values on snow depth applied in this study are sufficient to limit the impact of ground contamination on the snow albedo measurements. This allows us to have a robust comparison between simulated and observed snow albedo and to reduce the potential impact of ground contamination on the optimal ranges of gamma derived at each site. The results of the simulation with SNICAR will be added to the supplementary material. We will also mention these results in the revised paper when describing the criteria imposed to select days for the albedo evaluation:

*The first criterion ensured that the observed albedo corresponded to an actual snow albedo value. It made sure that the albedo measurement had been collected over a fully snow-covered ground and that the ground surface beneath the snow cover did not affect the measurement. Observed SD had to be higher than 20 cm. For high-latitude sites (above 60°N), this criterion was relaxed to 10 cm because there is less precipitation at these sites, so SD did not reach far over 20 cm in certain years. Albedo simulations with the radiative transfer model SNICAR v3 (Flanner et al., 2021) confirmed the ground surface beneath the snow cover does not modify the snow albedo by more than 2 % for a 10-cm thick snowpack (see Section xx of the supplementary material)*

We fully agree with Reviewer 2 that forest debris from surrounding high vegetation can influence the evolution of snow albedo in the visible range in forested terrain as shown by Melloh et al. (2001). Consequently, gamma should be modified in the presence of high vegetation to represent this effect. The present study focuses only on the large-scale variability of the snow darkening coefficient in open conditions (without trees) as a function of LAP deposition (BC and dust). For this reason, the three forested sites available in the ESM-SnowMIP dataset were not used in this study due to the potential impact of forest debris that could influence the optimal ranges of gamma at these sites as mentioned at L 136-138 in the submitted manuscript. The impact of forest debris on the values of gamma will be included in a new paragraph of the discussion:

*In addition, the effects of forest litter and debris on snow albedo in the visible range (Melloh et al., 2001) and on the resulting gamma values were not considered in this study. Only sites located in open terrain were selected to determine the optimal ranges of gamma. Therefore, the values of gamma shown on Fig. 12 do not include the effect of the forest presence. The approach of Hardy et al. (2000) could be considered to indirectly represent the effects of forest litter in the default snow albedo scheme used in Crocus.*

3. The climatology of LAP deposition will be a valuable dataset for global land surface models. However, the results presented in Fig. 12 have not been fully validated, and their accuracy, including associated uncertainties, remains unclear. Additionally, because the validation sites are limited, there are likely to be high uncertainties in the LAP data for regions such as South Asia, particularly in mountain glaciers. The authors should release the dataset only after thorough validation.

We agree with Reviewer 2 that the climatology of LAP deposition on snow and the associated values of the snow darkening coefficient are associated with uncertainties. To answer this comment, we will propose two additions to the revised paper:

1. An evaluation of the LAP deposition dataset
2. A quantification of the uncertainty in the values of the snow darkening coefficient resulting from the interannual variability of LAP deposition on snow.

These 2 major additions are described below.

1. A special section describing the accuracy of the GFDL LAP product will be added to the Supplementary material. It will contain a comparison between observed and simulated mean annual dust deposition at 26 sites around the world as shown in Figure 1. The GFDL product captures well the spatial variability of dust deposition around the globe. For example, it reproduces well, the contrasted dust deposition patterns in Central Asia (Fig 1b). For this evaluation, values from 6 different mountain glaciers in Asia have been considered (Table 1). The evaluation of BC climatological deposition is restricted to three sites at the moment as shown in Table 3. It shows that the LAP dataset captures the large differences in BC deposition between the Himalayan mountains and West Antarctica.

[Figure]

*Figure 1 (a) Map showing the mean annual dust deposition from the GFDL climatology (1981-2015), (b) same as (a) for a region centered around the Tibetan Plateau and (c) scatter plot comparing the annual dust deposition in the observation and in the GDFL climatology at 26 sites around the globe. The location of these sites is shown on maps (a) and (b). For each site, the pie chart shows the decomposition in the GFDL climatology between dry (orange) and wet (blue) deposition.*

Table 2: Mean annual dust deposition in the observations and in the GFDL climatology at 26 sites. The numbers correspond to the numbers shown on Fig 1.

| Num. | Name | Lat. | Lon. | Elev. (m) | Reference | Obs. (g m$^{-2}$yr$^{-1}$) | Clim. (g m$^{-2}$yr$^{-1}$) |
|---|---|---|---|---|---|---|---|
| 1 | Taklimakan | 39.75°N | 88.50°E | 200. | Zhang et al. (1998) | 450.00 | 177.37 |
| 2 | Waddan | 29.12°N | 15.93°E | 2. | O'Hara et al. (2006) | 74.53 | 34.61 |
| 3 | Tel Aviv | 32.00°N | 34.50°E | 40. | Ganor and Mamane (1982) | 36.00 | 26.87 |
| 4 | Ghana, Gulf of Guinea | 7.50°N | -1.50°E | 0. | Resch et al. (2008) | 22.00 | 9.66 |
| 5 | Erdemli | 33.57°N | 34.26°E | 21. | Kubilay et al. (2000) | 13.00 | 9.99 |
| 6 | Mera Ice Peak | 27.72°N | 86.87°E | 6376. | Ginot et al. (2014) | 10.40 | 11.39 |
| 7 | Karakoram Mtns | 36.00°N | 75.70°E | 5150. | Wake et al. (1994) | 9.60 | 11.28 |
| 8 | Pamir Mtns | 38.20°N | 75.10°E | 5910. | Wake et al. (1994) | 8.14 | 20.27 |
| 9 | Tuomuer Glacier | 41.75°N | 79.87°E | 4600. | Zhiwen and Zhongqin (2011) | 8.00 | 31.69 |
| 10 | Montseny Mtns | 41.80°N | 2.30°E | 700. | Avila et al. (1997) | 5.30 | 4.64 |
| 11 | So. Tibetan Plateau | 28.50°N | 87.50°E | 5850./6140. | Wake et al. (1994) | 3.39 | 5.28 |
| 12 | Tanggula Shan Mtns | 33.40°N | 91.10°E | 5950. | Wake et al. (1994) | 3.02 | 2.81 |
| 13 | French Alps | 45.50°N | 6.50°E | 4270. | Angelisi and Gaudichet (1991) | 2.10 | 3.81 |
| 14 | Miami | 25.75°N | -80.25°E | 10. | Prospero et al. (1987) | 1.62 | 1.10 |
| 15 | Tasman Glacier | -43.50°N | 170.30°E | 2600. | Windom (1969) | 1.20 | 0.82 |
| 16 | Renland | 71.30°N | -26.70°E | 2340. | Bory et al. (2003) | 0.68 | 0.73 |
| 17 | Midway | 28.20°N | -177.35°E | 4. | Prospero (1989) | 0.60 | 0.28 |
| 18 | Shemya | 59.92°N | 174.00°E | 2. | Prospero (1989) | 0.60 | 0.65 |
| 19 | Navarino Island | -55.22°N | -67.62°E | 35. | Sapkota et al. (2007) | 0.43 | 0.53 |
| 20 | Oahu | 21.30°N | -157.60°E | 20. | Prospero (1989) | 0.42 | 0.13 |
| 21 | Mount Olympus | 47.00°N | -123.00°E | 2000. | Windom (1969) | 0.32 | 0.47 |
| 22 | New Zealand | -34.50°N | 172.75°E | 2. | Arimoto et al. (1990) | 0.14 | 0.41 |
| 23 | Fanning | 3.90°N | -159.30°E | 25. | Prospero (1989) | 0.05 | 0.04 |
| 24 | James Ross Island | -64.20°N | -57.70°E | 1542. | McConnell et al. (2007) | 0.03 | 0.20 |
| 25 | GRIP | 72.60°N | -37.60°E | 3232. | Bory et al. (2003) | 0.02 | 0.18 |
| 26 | Enewetak | 11.30°N | 162.30°E | 5. | Arimoto et al. (1985) | 0.02 | 0.04 |

Table 3: Mean annual black carbon deposition in the observations and in the GFDL climatology at three sites.

| Num. | Name | Lat. | Lon. | Elev. (m) | Reference | Obs. (g m$^{-2}$yr$^{-1}$) | Clim. (g m$^{-2}$yr$^{-1}$) |
|---|---|---|---|---|---|---|---|
| 1 | Nepal Clim. Obs.- Pyramid | 27.95°N | 86.80°E | 5079. | Yasunari et al. (2010) | 112.00 | 179.83 |
| 2 | Tibet Palong-Zanbu Glacier | 29.21°N | 96.92°E | 5500. | Xu et al. (2009) | 29.40 | 77.40 |
| 3 | West Antarctic Ice Sheet | -79.46°N | -112.08°E | 1766. | Bisiaux et al. (2012) | 0.02 | 0.07 |

2. The impact of the interannual variability of LAP deposition will be considered in the revised version of the manuscript. The GFLD dataset consists of different percentiles (1, 5, 10, 25, 50, 75, 90, 95, 99) of BC and dust deposition rates over the period 1979-2015, for each day of the year. In the submitted paper, the median deposition rate for each day has been considered and combined with the GMASI daily snow cover climatology to obtain an estimation of the mean daily LAP deposition on snow. In

the revised manuscript, we will consider the percentiles 25 and 75 for dust and BC to obtain an estimation of typical daily average LAP deposition on snow for a low LAP year and for a high LAP year. This method provides an estimation of the interannual variability of LAP deposition on snow. The limitations associated with this estimation are mentioned in a paragraph below.

The daily average deposition of dust and BC on snow for a low (high) LAP year will be computed by replacing the median deposition rate by the percentile 25 (75) for each day of the year in Eq. 5 of the submitted paper. The daily average deposition of LAP on snow for a low (high) LAP will be finally expressed in equivalent BC using Eq. 6 of the submitted paper. A coefficient of variability, $C_{v,D}$, will be computed to characterize the inter-annual variability of LAP deposition on snow with respect to the climatology:

$$C_{v,D} = (D_{high} - D_{low})/D_{mean}$$

Where $D_{high}$, $D_{low}$ and $D_{mean}$ corresponds to the daily average LAP deposition on snow for a high LAP year, a low LAP year and the climatology. Figures 2 and 3 below show how the interannual range LAP deposition on snow $(D_{high} - D_{low})$ and the coefficient of variability, $C_{v,D}$, vary in space. Figure 2 will be added to the supplementary material of the revised paper whereas Figure 3 will be added to the main revised manuscript.

[Figure]

Figure 2: Global maps of the interannual range of daily (a) BC, (b) dust, and (c) total LAP (BC + dust in equivalent BC) deposition rates over snow between a high and a low LAP year $(D_{high} - D_{low})$.

[Figure]

*Figure 3: Global maps of the coefficient of variability of daily (a) BC, (b) dust, and (c) total LAP (BC + dust in equivalent BC) deposition rates over snow between a high and a low LAP year.*

The LAP deposition rates for a low and a high LAP year will be shown in the cross analysis to illustrate the interannual variability of LAP deposition of snow at each of the experimental sites considered in the study. Figure 9 in the submitted manuscript will be updated to show the interannual variability at all the different sites and the effect of the altitudinal correction (see Fig. 4 below for the revised figure). This figure shows that, despite the interannual variability, sites such as Sapporo or Col de Porte receive significantly larger amounts of LAP on snow than Arctic sites such as Bylot and Umiujaq. The regression between the optimal values of gamma and the climatological LAP deposition on snow at each of the sites won't be modified in the revised manuscript. Indeed, the optimal ranges of gamma at a given site have been selected based on the median RMSE of simulated snow albedo over all the years for each value of gamma. This range is therefore related to the mean climatological LAP deposition on snow at each of the sites. Figure 10 in the submitted manuscript showing the linear regression between gamma and LAP will be updated to show the interannual variability of LAP deposition on snow at each site  (see Fig. 4 below for the revised figure).

[Figure]

Figure 4: Graph representing climatological LAP deposition rates at all sites, before and after the correction of elevation effects. Arrows indicate these corrections at mountainous sites. The error bar represents the interannual variability of LAP deposition at each site.

Figure 5: Graph representing the optimal ranges of γ as a function of total LAP deposition rates in equivalent BC, in logarithmic scales, and the corresponding linear regressions with and without correction for elevation at the mountain sites (the correction is represented by the arrows). The error bar represents the interannual variability of LAP deposition at each site.

The estimation of the interannual variability of LAP deposition on snow is calculated for a low LAP year and a high LAP year using the percentiles 25 and 75 for dust and BC deposition available in the GFDL dataset. Such a method can only provide an estimation of interannual variability. A more accurate computation would have required to use the deposition rates of BC and dust for each day of the period 1979-2015 in the GDFL dataset and to combine them with the corresponding GMASI snow cover information. Mean daily deposition of LAP on snow could have then been computed for each year of the climatology and the interannual variability could have been estimated precisely. Unfortunately, the daily LAP deposition data were not available in the version of the GFDL dataset used in this study. Future work could use these daily data once they are available or use another source of global LAP deposition data at daily time scale such as the MERRA-2 dataset. These limitations will be mentioned in the discussion of the revised manuscript.

The values $D_{high}$ and $D_{low}$ for dust, BC and total LAP will be added to the dataset distributed on Zenodo. This will provide the users of the climatology of LAP deposition on snow with an estimation of the interannual variability of LAP deposition on snow.

The uncertainty in the estimation of gamma resulting from the interannual variability of LAP deposition on snow will be quantified in the revised paper. We propose to account for this uncertainty in the revised paper by computing the values of gamma corresponding to a low LAP year and to a high LAP year using the regression given in Eq. 9 of the original paper and the map of LAP deposition on snow for a low LAP and high LAP year. Similarly to the coefficient of variability computed for the LAP deposition on snow, we propose to compute a coefficient of variability for the value of gamma:

$$C_{v,\gamma} = (\gamma_{high} - \gamma_{low})/\gamma_{mean}$$

Figure 12 in the submitted paper will be updated to add a map that shows how $C_{v,\gamma}$ varies globally (see Fig. 6 below). The largest uncertainties for gamma are found in regions where the interannual variability of LAP deposition on snow is the largest. This includes the east coast of North America and central Siberia that are close to regions of BC emissions. The Karakum Desert, the Aral Sea and the Gobi Desert are also regions of large variability in gamma due to the strong interannual variability of dust deposition in these regions. Finally, Antarctica shows no interannual variability of gamma since the gamma values for the low and high years are equal to 900 days (the maximal value allowed for gamma in Crocus) due to very low LAP deposition on snow.

[Figure]

*Figure 6: Global maps showing (a) the optimal value of γ derived and (b) the associated uncertainty due to the LAP interannual variability.*

The values $\gamma_{high}$ and $\gamma_{low}$ will be added to the dataset distributed on Zenodo. This will allow the users of the model to know where the values of gamma are uncertain due to the interannual variability of LAP deposition on snow. Such uncertainty could be considered in the context of ensemble snowpack simulations.

(minor comments)

1. It is understood that Crocus is a multi-layer model used to simulate snow conditions across various layers. However, only two snow grain size parameters (d_opt and d') are employed to calculate the albedo in the visible and near-infrared regions (Table 1). In the visible range, ice exhibits weak light absorption, allowing light to penetrate more deeply. Therefore, to accurately calculate the albedo in the visible range, it is crucial to consider the vertical distribution of snow grain sizes. Please add an explanation of which snowpack layer d_opt parameter represents.

Thanks for this comment. We agree with Reviewer 2 that the description of how Crocus handles solar radiation was not sufficient in the submitted manuscript. We will add the following information about the model. First, the snow albedo in Crocus is computed as a depth-weighted average of the albedo of the two upper snow layers, so on a total depth of a few to a tens of cm. The properties (d_opt, age) of each layer are used to compute their respective albedo. This method is applied to avoid time discontinuities in the simulated snow albedo resulting from layer aggregation. Second, Crocus simulates the penetration of incoming shortwave radiation into the snowpack and its absorption assuming an exponential decay of radiation with increasing snow depth. The absorption coefficient in the different spectral bands depends on the optical diameter and on the density of each snow layer.

The following sentences will be added to the revised paper:

*The snow albedo in each band is obtained as a depth-weighted average of the albedo of the two upper snow layers. The properties (d_opt, age) of each layer are used to compute their respective albedo. This method is applied to avoid time discontinuities in the simulated snow albedo resulting from layer aggregation.*

*Crocus simulates the penetration of incoming shortwave radiation into the snowpack and its absorption assuming an exponential decay of radiation with increasing snow depth. The absorption coefficient in the different spectral bands (Table 1) depends on the optical diameter and on the density of each snow layer.*

Table 1 in the paper will be revised to include the snow absorption coefficient for each band as shown below:

**Table 1.** Equations representing the snow albedo and the snow absorption coefficient for the three spectral bands in Crocus. The parameters are as follows: $d_{opt}$ (m) is the optical grain diameter of the snow, $\rho$ the snow density, $P$ (Pa) is the mean pressure at the site, $P_{CDP}$ (Pa) is the mean pressure at the Col de Porte site, $A$ (days) is the age of the snow, and $\gamma$ (days) is the snow  darkening coefficient. Adapted from Table 4 in Vionnet et al. (2012).

| Spectral band | Spectral albedo $\alpha_{spectral\ band}$ | Absorption coefficient $\beta$ (m$^{-1}$) |
|---|---|---|
| 0.3—0.8 μm | $\alpha_{0.3-0.8\mu m} = \max(0.6, \alpha_i - \Delta\alpha_{age})$ $where : \alpha_i = \min(0.92, 0.96 - 1.58\sqrt{d_{opt}})$ $and : \Delta\alpha_{age} = \min(1, \max(\frac{P}{P_{CDP}}, 0.5)) \times 0.2\frac{A}{\gamma}$ | $\beta_{0.3-0.8\mu m} = \max(40, 0.00192 \frac{\rho}{\sqrt{d_{opt}}})$ |
| 0.8—1.5 μm | $\alpha_{0.8-1.5\mu m} = \max(0.3, 0.9 - 15.4\sqrt{d_{opt}})$ | $\beta_{0.8-1.5\mu m} = \max(100, 0.01098 \frac{\rho}{\sqrt{d_{opt}}})$ |
| 1.5—2.8 μm | $\alpha_{1.5-2.8\mu m} = 346.3d' - 32.31\sqrt{d'} + 0.88$ $where : d' = \min(d_{opt}, 0.0023)$ | $\beta_{1.5-2.8\mu m} = +\infty$ |

2. Please explain how the solar zenith angle dependence of snow albedo and the effects of the atmosphere, especially clouds, on snow albedo can be expressed in terms of Eq. 1.

The dependence of snow albedo on the solar zenith angle and the influence of the cloud cover and optical thickness on the distribution of the incoming radiation in the three spectral bands is not considered in the default albedo scheme in Crocus. This limitation will be explicitly mentioned in the revised manuscript when detailing the snow albedo parameterization in Crocus:

*The model does not include the change in the ratio of incoming radiation in the three bands as a function of the sky conditions and of the solar zenith angle. The impact of these limitations is discussed in Sect. 5.*

We will then discuss the uncertainties associated with these limitations in the discussion of the revised paper:

*Despite the improvements brought in this study, the default snow albedo in Crocus still suffers from limitations associated with the fixed ratio used to compute the broadband albedo from the values in the three spectral bands (Eq. 1) and the absence of effect of the solar zenith angle on snow albedo. Gardner and Sharp (2010) have shown that changes in solar zenith angle and clouds' optical*

*thickness can lead to changes on the order of 0.05 in the snow albedo values (Fig. 9 in their study). Including these effects will lead to further improvements in large scale snow albedo simulations with Crocus.*

3. For reference, please also show the results for SSA=10 m^2kg^-1 (~granular snow) in Fig. 1.

The two figures below show how the snow albedo decrease with snow age varies as a function of the snow darkening coefficient gamma for two values of SSA (50 $m^2$ $kg^{-1}$ as in the submitted paper and 10 $m^2$ $kg^{-1}$ as suggested by Reviewer 2). In the visible range (Figure 2), the two graphics are identical since the reference albedo value for a snow age of 0 is equal to 0.92 for both values of SSA (see Table 1 in the main manuscript). On the other hand, the two SSA values lead to different albedo values in the two near-infrared bands so that for a given age and a given value of gamma, the broadband albedo is lower for SSA = 10 $m^2$ $kg^{-1}$ than for SSA = 50 $m^2$ $kg^{-1}$.

We have decided to keep unchanged Figure 1 in the revised manuscript since the two values of SSA give the same albedo evolution as a function of snow age in the visible range.

[Figure]

*Figure 6 Graphic representation of the dependency of the snow albedo in the visible spectral band (0.3–0.8 µm) to the snow age for different values of gamma and for two SSA values (50 m2/kg and 10 m2/kg)*

[Figure]

*Figure 7 Graphic representation of the dependency of the broadband snow albedo (0.3−0.8μm) to the snow age for different values of gamma and for two SSA values (50 m2/kg and 10 m2/kg)*

---

## Author Response (AR1)

**Answer to Reviewer 1**

We thank Reviewer 1 for their comments. We provide here our responses to those comments and describe how we addressed them in the revised manuscript. The original reviewer comments are in normal black font while our answers appear in blue font. The line numbers mentioned below correspond to the line numbers in the manuscript in track-change mode.

Light-absorbing particles (LAPs, e.g., BC and dust) deposited at the snow surface significantly reduce snow albedo and strongly affect the snow dynamics. This study analyzed the relationship between LAP concentration and the optimized parameter that controls the evolution of snow albedo in Crocus model. However, there are limited sites in the study and cannot be used to upscale the impacts from site to globe. The spatial match between field measurements and gridded snow cover and LAP data is not well evaluated. The global simulations are not well evaluated and compared. The authors used climatological yearly-average data to analyze the relationship and neglected the seasonal variability and interannual variability of LAP concentration. Please see below for my specific comments.

**Major concerns:**

1. Line 45: The authors stated that some snowpack models in LSMs often rely on empirical parameterizations. However, some widely-used LSMs, e.g., CLM and ELM, use a mechanism-based SNICAR model to simulate snow albedo with the consideration of LAP impacts. Indeed, they need the input of LAP deposition fluxes. Please state the limitation of such models.

    Need to reformulate in the intro should be OK.

The introduction has been revised to explicitly mention that radiative transfer models are used to simulate snow albedo evolution in land surface models such as CLM and ELM. The following sentences were added at the end of the third paragraph of the introduction (L 47-50):

*These coupled models are also used in the context of land and climate models such as in the Community Land Model (Lawrence et al., 2019) and in the land component of the Energy Exascale Earth System Model (Golaz et al., 2019). He et al. (2024) have demonstrated how such models can improve global simulations of snowpack evolution with a positive impact on the simulations of near-surface temperature.*

We have also carried out a complementary analysis of the snow albedo parameterizations used in the 21 snowpack models that have been involved in the ESM-Snow MIP project (Menard et al., 2019). Among those models, only two models (CLM and SMAP) used a snow albedo scheme that relies on a snow radiative transfer model that explicitly considers the effects of LAP on snow albedo. This analysis reveals that even if the importance of LAP on snow albedo evolution is well known, many models still rely on parameterizations of various complexity that aim at achieving high computational efficiency with different levels of physical realism. This highlights the importance of pursuing improvements of such parameterizations as proposed by our study. The following sentence has been added to the introduction of the revised manuscript (L 57-62):

*For example, among the 21 snowpack models used in the Earth System Model - Snow Model Intercomparison Project (ESM-Snow MIP, Menard et al., 2019), only two of them (CLM, Lawrence et al., 2019; and the Snow Metamorphism and Albedo Process model (SMAP), Niwano et al., 2012) explicitly represent the effects of LAP on snow albedo evolution. The other models rely on parameterizations of varying complexity (Supplementary material of Menard et al., 2019) that aim at achieving high computational efficiency with different levels of physical realism.*

The limitations associated with the combination of a snow radiative transfer model with a snow model were already mentioned in the submitted manuscript (L45-49). They consist in the additional computational costs and the spectral resolution of the radiative transfer model and the need for additional atmospheric forcings (LAP deposition fluxes) to drive the snow model. These lines remained unchanged in the revised manuscript.

2. Line 54: The authors stated that the parameterizations use fixed time constants and were optimized using observation data. Looks like this study also used limited field data to optimize the models, which cannot ensure the accuracy at the untested grids. Please clearly explain it.

The sentence mentioned by Reviewer 1 is taken from the introduction in and reads as:

*These parameterizations use fixed time constants that have been optimized using sparse observational data, restricting their extension to untested environments and time periods (Molders et al., 2008).*

This sentence refers to the simple time/temperature dependent parameterizations that are still used in many snow models (8 models among the 21 models tested in the context of ESM Snow MIP; Supplement of Menard et. al., 2019). In parameterizations such as the one developed by Douville et al. (1995), two time constants (one for dry snow, one for wet snow) are used to simulate the decrease of snow albedo with time due to multiple snow evolution processes (increase in grain size due to snow metamorphism, LAP deposition, ...). The same constants are applied in any points of the globe when such models are applied globally. These constants have often been set using a restricted set of observations. For example, the weather-dependent snow albedo parameterization of Molders et al (2008) mentioned in the introduction has been developed using data from two winters at one site in Alaska.

In the approach presented in this study, multi-year data from 10 sites covering different snow classes around the globe are used to derive optimal ranges of the snow darkening coefficient (snow aging coefficient in the submitted paper, see answer to comment 4) for the default snow albedo parameterization in Crocus. A total of 121 winters of snow albedo measurements are used for this evaluation. We agree with Reviewer 1 that this evaluation is still limited, but we believe it provides more robustness than previous evaluations of snow albedo parameterizations. The main challenge to increase the robustness is to add new sites with quality-controlled meteorology and dedicated snow observations as mentioned in the discussion of the submitted paper (lines 435-437).

3. Table 1: Please provide more information on the empirical equations. How did the authors set the values in the equations.

The equations presented in Table 1 are the equations from the reference snow albedo scheme in Crocus. The values in the equations have not been set in the context of this study but in the context of the original development of the model (Brun et al., 1992). The empirical parameters that relate the snow albedo with the snow optical diameter have been derived from Warren (1982) and Sergent et al. (1987). These two references were added to the revised manuscript (L 108-109):

*The albedo in each spectral band is calculated using a different equation (Table 1) as proposed by Brun et al. (1992) based on the work from Warren (1982) and Sergent et al. (1987).*

The empirical relationship between snow albedo and snow age through the snow darkening coefficient has also been proposed by Brun et al (1992) and tuned for the Col de Porte experimental site in the French Alps with the time series available at that time (of course much shorter than today) as already explained in the initial manuscript (L 104-106 of the submitted document).

4. Equation 1: the ratio of incoming radiation at the three bands can vary with sky conditions. The fixed ratio in this equation can induce some uncertainties. Please evaluate it and discuss it.

We fully agree with Reviewer 1 that the distribution of incoming radiation in the three bands can vary as a function of sky conditions (e.g. cloudiness) and with the solar zenithal angle. This limitation is explicitly mentioned in the revised manuscript when detailing the snow albedo parameterization in Crocus (L 134-136):

*The model does not include the change in the ratio of incoming radiation in the three bands as a function of the sky conditions and of the solar zenith angle. The impact of these limitations is discussed in Sect. 5.*

We then discuss the uncertainties associated with these limitations in the discussion of the revised paper (L 541-545):

*Despite the improvements brought in this study, the default snow albedo in Crocus still suffers from limitations associated with the fixed ratio used to compute the broadband albedo from the values in the three spectral bands (Eq. 1) and the absence of effect of the solar zenith angle on snow albedo. Gardner and Sharp (2010) have shown that changes in solar zenith angle and clouds' optical thickness can lead to changes on the order of 0.05 in the snow albedo values (Fig. 9 in their study). Including these effects will lead to further improvements in large scale snow albedo simulations with Crocus.*

5. In my view, snow age should be related to snow grain size. Please explain the relationship between d_opt and snow aging coefficient. I am not sure whether calling gamma as snow aging coefficient is reasonable or not. How did the authors model snow age?

In Crocus, the snow age simply corresponds to the time in days since the last snowfall for a given snow layer. A new layer made of fresh snow is given a snow age of 0 and the snow age increases with time if no snowfall occurs. When two layers are merged, the resulting snow age is the mass-weighted average of the snow ages of the two respective layers. A definition of the snow age in Crocus has been added to the revised paper (L 124-126):

*The snow age in Crocus corresponds to the time in days since the last snowfall for a given snow layer. A new layer made of fresh snow is given a snow age of 0 and the snow age increases with time if no snowfall occurs.*

The evolution of the optical grain size in Crocus is computed using metamorphism laws as described in Brun et al. (1992) and Carmagnola et al. (2014). For dry snow, the temporal evolution of the optical grain size is a function of the temperature gradient between the different layers. For wet snow, the increase in optical grain size with time depends on the liquid water content in the different layers. Therefore, the optical grain size in Crocus is not directly related to the snow age. Information about the evolution of the optical grain size in Crocus were added to Section 2 of the revised paper (L 108-112):

*The evolution of the optical grain size in Crocus is computed using metamorphism laws as described in Brun et al. (1992) and Carmagnola et al. (2014). For a given layer, for dry snow, the temporal evolution of the optical grain size is a function of the vertical temperature gradient, whereas, for wet snow, the increase in optical grain size with time depends on the snow liquid water content.*

We agree with Reviewer 1 that the term "snow aging coefficient" was creating confusion for the reader since this coefficient is used to only empirically represent the impact of the deposition of LAP on the snow albedo in the visible band. It does not represent all the snow aging processes impacting the snow albedo evolution such as the change in optical grain size due to metamorphism, because this feedback is already explicitly simulated in the model thanks to the metamorphism parameterizations and the relationship between optical diameter and albedo in Table 1. For this reason, gamma will be renamed the "snow darkening coefficient" in the revised manuscript. We believe this new name will help the reader to better understand the role of gamma in the evolution of snow albedo in Crocus.

6. Section 3.2.1: Please briefly introduce the accuracy of two products: LAP and snow cover.

A special section describing the accuracy of the GFDL LAP product has been added to the Supplementary material (Section 1.2). It contains a comparison between observed and simulated mean annual dust deposition at 26 sites around the world as below shown on Figure 1 and detailed in Table 1. The GFDL product captures well the spatial variability of dust deposition around the globe. For example, it reproduces well the contrasted dust deposition patterns in Central Asia (Fig 1b). The evaluation of BC climatological deposition is restricted to three sites at the moment as shown in Table 2. It shows that the LAP dataset captures the large differences in BC deposition between the Himalayan mountains and West Antarctica.

[Figure]

*Figure 1 (a) Map showing the mean annual dust deposition from the GFDL climatology (1981-2015), (b) same as (a) for a region centered around the Tibetan Plateau and (c) scatter plot comparing the annual dust deposition in the observation and in the GDFL climatology at 26 sites around the globe. The location of these sites is shown on maps (a) and (b). For each site, the pie chart shows the decomposition in the GFDL climatology between dry (orange) and wet (blue) deposition.*

Table 1: Mean annual dust deposition in the observations and in the GFDL climatology at 26 sites. The numbers correspond to the numbers shown on Fig 1.

| Num. | Name | Lat. | Lon. | Elev. (m) | Reference | Obs. (g m$^{-2}$yr$^{-1}$) | Clim. (g m$^{-2}$yr$^{-1}$) |
|---|---|---|---|---|---|---|---|
| 1 | Taklimakan | 39.75°N | 88.50°E | 200. | Zhang et al. (1998) | 450.00 | 177.37 |
| 2 | Waddan | 29.12°N | 15.93°E | 2. | O'Hara et al. (2006) | 74.53 | 34.61 |
| 3 | Tel Aviv | 32.00°N | 34.50°E | 40. | Ganor and Mamane (1982) | 36.00 | 26.87 |
| 4 | Ghana, Gulf of Guinea | 7.50°N | -1.50°E | 0. | Resch et al. (2008) | 22.00 | 9.66 |
| 5 | Erdemli | 33.57°N | 34.26°E | 21. | Kubilay et al. (2000) | 13.00 | 9.99 |
| 6 | Mera Ice Peak | 27.72°N | 86.87°E | 6376. | Ginot et al. (2014) | 10.40 | 11.39 |
| 7 | Karakoram Mtns | 36.00°N | 75.70°E | 5150. | Wake et al. (1994) | 9.60 | 11.28 |
| 8 | Pamir Mtns | 38.20°N | 75.10°E | 5910. | Wake et al. (1994) | 8.14 | 20.27 |
| 9 | Tuomuer Glacier | 41.75°N | 79.87°E | 4600. | Zhiwen and Zhongqin (2011) | 8.00 | 31.69 |
| 10 | Montseny Mtns | 41.80°N | 2.30°E | 700. | Avila et al. (1997) | 5.30 | 4.64 |
| 11 | So. Tibetan Plateau | 28.50°N | 87.50°E | 5850./6140. | Wake et al. (1994) | 3.39 | 5.28 |
| 12 | Tanggula Shan Mtns | 33.40°N | 91.10°E | 5950. | Wake et al. (1994) | 3.02 | 2.81 |
| 13 | French Alps | 45.50°N | 6.50°E | 4270. | Angelisi and Gaudichet (1991) | 2.10 | 3.81 |
| 14 | Miami | 25.75°N | -80.25°E | 10. | Prospero et al. (1987) | 1.62 | 1.10 |
| 15 | Tasman Glacier | -43.50°N | 170.30°E | 2600. | Windom (1969) | 1.20 | 0.82 |
| 16 | Renland | 71.30°N | -26.70°E | 2340. | Bory et al. (2003) | 0.68 | 0.73 |
| 17 | Midway | 28.20°N | -177.35°E | 4. | Prospero (1989) | 0.60 | 0.28 |
| 18 | Shemya | 59.92°N | 174.00°E | 2. | Prospero (1989) | 0.60 | 0.65 |
| 19 | Navarino Island | -55.22°N | -67.62°E | 35. | Sapkota et al. (2007) | 0.43 | 0.53 |
| 20 | Oahu | 21.30°N | -157.60°E | 20. | Prospero (1989) | 0.42 | 0.13 |
| 21 | Mount Olympus | 47.00°N | -123.00°E | 2000. | Windom (1969) | 0.32 | 0.47 |
| 22 | New Zealand | -34.50°N | 172.75°E | 2. | Arimoto et al. (1990) | 0.14 | 0.41 |
| 23 | Fanning | 3.90°N | -159.30°E | 25. | Prospero (1989) | 0.05 | 0.04 |
| 24 | James Ross Island | -64.20°N | -57.70°E | 1542. | McConnell et al. (2007) | 0.03 | 0.20 |
| 25 | GRIP | 72.60°N | -37.60°E | 3232. | Bory et al. (2003) | 0.02 | 0.18 |
| 26 | Enewetak | 11.30°N | 162.30°E | 5. | Arimoto et al. (1985) | 0.02 | 0.04 |

Table 2: Mean annual black carbon deposition in the observations and in the GFDL climatology at three sites.

| Num. | Name | Lat. | Lon. | Elev. (m) | Reference | Obs. (g m$^{-2}$yr$^{-1}$) | Clim. (g m$^{-2}$yr$^{-1}$) |
|---|---|---|---|---|---|---|---|
| 1 | Nepal Clim. Obs.- Pyramid | 27.95°N | 86.80°E | 5079. | Yasunari et al. (2010) | 112.00 | 179.83 |
| 2 | Tibet Palong-Zanbu Glacier | 29.21°N | 96.92°E | 5500. | Xu et al. (2009) | 29.40 | 77.40 |
| 3 | West Antarctic Ice Sheet | -79.46°N | -112.08°E | 1766. | Bisiaux et al. (2012) | 0.02 | 0.07 |

This new section in the supplementary material is mentioned in Section 3.2.1 of the main manuscript (L 268-270):

*A complementary analysis of the accuracy of the GFDL dataset is presented in Sect. 1.2 of the Supplementary material. In particular, it shows that the spatial variability of the mean annual dust deposition is well captured by the GDFL dataset (Fig. S3 in the Supplementary material).*

The following information about the GMASI snow product has been added to the revised manuscript (L 280-285):

*The accuracy of the GMASI snow product has been evaluated by Romanov (2017) versus in situ and other remote sensing datasets. Over the continental United States, he found that the percentage of correct snow identifications when compared with in situ observations ranged between 75 and 85 % for the winter months and 94 % over the full year. When compared with the Interactive Multisensor Snow and Ice Mapping System (Helfrich et al., 2007) over the whole land area of the Northern Hemisphere, the agreement remained above 90 % all year long. The largest differences between the two products were found during the fall and the spring due to fast changes in the snow cover.*

7. The authors neglected the impacts of brown carbon. Please at least discuss the potential uncertainty.

A new paragraph has been added to the discussion in the revised paper to highlight the factors that are influencing the values of gamma and that have been neglected in this study. This includes brown carbon as highlighted by Reviewer 1. The beginning of this paragraph (L 563-569) reads as:

*This study has established a relationship between the snow darkening coefficient (gamma) and the climatological values of LAP deposition on snow. Several factors that are influencing the spatial distribution of gamma are not included. Firstly, only deposition of dust and BC were considered when computing the climatology of mean LAP deposition over snow. The darkening effect of brown carbon (BrC) resulting from biomass burning and biofuel sources (Beres et al., 2020; Brown et al., 2022) is not included since BrC deposition rates are not available in the GFDL dataset. Brown et al. (2022) found that the ratio of globally averaged snow darkening effect from BrC to the one from BC ranges from 37% to 98%. This suggests that including BrC could help refine the global map of gamma shown on Fig. 12.*

8. The authors matched the field measurements (snow albedo) and coarse simulated gridded data. This process may cause large uncertainties. Please evaluate the site spatial representativeness before do such spatial matching.

In our study, the snow albedo measured in the field has been compared with simulated snow albedo derived from point scale simulations with the snowpack scheme Crocus. These simulations are driven by local observed meteorological forcing so that they are representative of the local scale conditions encountered at each site. Therefore, the optimal values of the snow darkening coefficient, gamma, that have been obtained at each site do not suffer from a limitation associated with site spatial representativeness. We have added a sentence in Section 3.1.1 of the revised manuscript (L 183-184) to insist on the local character of the snowpack simulations:

*At each site, the atmospheric forcing is representative of the local meteorological conditions and is mostly made of observations (Menard et al., 2019).*

The question of site spatial representativeness arises when comparing the optimal values of gamma at each site with the LAP climatological mean deposition at each site. Since these values are not observed at the sites, they have been taken from a coarse simulated gridded product (the GDFL LAP dataset in our case). Extracting LAP deposition rates from coarse gridded products is often employed when driving snowpack simulations at local sites to study the impact of LAP on snow dynamics (e.g.

Tuzet et al., 2020; Zorzetto et al., 2024). In these studies, the information from the closest grid point is often used without any adjustment. In our study, for each site, a correction based on the local altitudinal gradient of LAP deposition has been applied to consider the elevation difference between the actual elevation of the site and the elevation of the GFDL dataset at the location of the site. This correction aims at improving the spatial representativeness of the coarse gridded LAP deposition dataset at each site. Nonetheless, the GFDL climate model at 50-km grid spacing cannot accurately represent local processes influencing LAP deposition in mountainous terrain as mentioned in L 411-413 in the original paper.

9.  As I know, the interannual variability of LAP deposition is very large. However, the authors used the climatological data in the study. Please evaluate whether such use is reasonable.

We thank Reviewer 1 for this comment. The impact of the interannual variability of LAP deposition is now considered in the revised version of the manuscript. The GFLD dataset consists of different percentiles (1, 5, 10, 25, 50, 75, 90, 95, 99) of BC and dust deposition rates over the period 1979-2015, for each day of the year. In the submitted paper, the median deposition rate for each day was considered and combined with the GMASI daily snow cover climatology to obtain an estimation of the mean daily LAP deposition on snow. In the revised manuscript, we have considered the percentiles 25 and 75 for dust and BC to obtain an estimation of typical daily average LAP deposition on snow for a low LAP year and for a high LAP year. This method provides an estimation of the interannual variability of LAP deposition on snow.

The daily average deposition of dust and BC on snow for a low (high) LAP year has been computed by replacing the median deposition rate by the percentile 25 (75) for each day of the year in Eq. 5 of the submitted paper. The daily average deposition of LAP on snow for a low (high) LAP has been finally expressed in equivalent BC using Eq. 6 of the submitted paper. A coefficient of variability, $C_{v,D}$, has been computed to characterize the inter-annual variability of LAP deposition on snow with respect to the climatology:

$$C_{v,D} = (D_{high} - D_{low})/D_{mean}$$

Where $D_{high}$, $D_{low}$ and $D_{mean}$ corresponds to the daily average LAP deposition on snow for a high LAP year, a low LAP year and the climatology. Figure 2 and 3 below show how the interannual range LAP deposition on snow ($D_{high} - D_{low}$) and the coefficient of variability, $C_{v,D}$, vary in space. Figure 2 has been added to the supplementary material of the revised paper (Figure S.3) whereas Figure 3 has been added to the main revised manuscript (Figure 8 in the revised document))

[Figure]

Figure 2: Global maps of the interannual range of daily (a) BC, (b) dust, and (c) total LAP (BC + dust in equivalent BC) deposition rates over snow between a high and a low LAP year $(D_{high} - D_{low})$.

[Figure]

*Figure 3: Global maps of the coefficient of variability of annual mean (a) BC, (b) dust, and (c) total LAP (BC + dust in equivalent BC) deposition rates over snow between a high and a low LAP year.*

A new paragraph has been added to the results section to describe Figure 3 (L 398-408):

*Figure 8 quantifies the inter-annual variability of BC, dust and total LAP deposition on snow. The Mediterranean basin, Turkey, the Caucasus, the southern Andes and the mountains of Western US presents a large inter-annual variability of dust deposition on snow associated with the strong variability of dust deposition events in these regions (Skiles et al., 2012; Di Mauro et al., 2015; Dumont et al., 2020; Reveillet et al., 2022; Haugvaldstad et al., 2024). The inter-annual variability of dust deposition on snow is generally larger than the inter-annual variability of BC deposition on snow, in agreement with (Reveillet et al., 2022). Antarctica appears as the region with the strongest inter-annual variability of BC deposition on snow (compared to the climatology) but this variability remains the lowest around the planet when considered in absolute value (Fig. S4 in the Supplementary Material). As for the total deposition rates, the inter-annual variability of total LAP deposition on snow remains close to the values found for BC, at the exception of regions where the inter-annual variability of dust deposition on snow is the largest (Karakum Desert, Aral Sea, Gobi Desert and south-western Andes).*

The LAP deposition rates for a low and a high LAP year are shown in the cross analysis to illustrate the interannual variability of LAP deposition of snow at each of the experimental sites considered in the study. Figure 9 in the submitted manuscript has been updated to show the interannual variability at all the different sites and the effect of the altitudinal correction (see Fig. 4 below for the revised figure). This figure shows that, despite the interannual variability, sites such as Sapporo or Col de Porte receive significantly larger amounts of LAP on snow than Arctic sites such as Bylot and Umiujaq. The regression between the optimal values of gamma and the climatological LAP deposition on snow at each of the sites were not modified in the revised manuscript. Indeed, the optimal ranges of gamma at a given site have been selected based on the median RMSE of simulated snow albedo over all the years for each value of gamma. This range is therefore related to the mean climatological LAP deposition on snow at each of the sites. Figure 10 in the submitted manuscript showing the linear regression between gamma and LAP has also been updated to show the interannual variability of LAP deposition on snow at each site (see Fig. 5 below for the revised figure).

[Figure]

*Figure 4: Graph representing climatological LAP deposition rates at all sites, before and after the correction of elevation effects. Arrows indicate these corrections at mountainous sites. The error bar represents the interannual variability of LAP deposition at each site.*

[Figure]

*Figure 5: Graph representing the optimal ranges of γ as a function of total LAP deposition rates in equivalent BC, in logarithmic scales, and the corresponding linear regressions with and without correction for elevation at the mountain sites (the correction is represented by the arrows). The error bar along the x-axis represents the interannual variability of LAP deposition at each site.*

The estimation of the interannual variability of LAP deposition on snow is calculated for a low LAP year and a high LAP year using the percentiles 25 and 75 for dust and BC deposition available in the GFDL dataset. The limitations associated with this approach are mentioned in the discussion (L 509 525):

*The LAP deposition outputs in the GFDL dataset only contain the deposition rates for BC and dust as percentiles for each day of the year, based on the 1979-2015 period. This characteristic of the dataset led to two limitations of this study. First, the inter-annual variability of LAP deposition on snow was calculated for a low LAP year and a high LAP year using the percentiles 25 and 75 for dust and BC deposition for each day of the year. Such a method can only provide an estimation of the inter-annual variability of LAP deposition. A more accurate computation would have required to use the deposition rates of BC and dust for each day of the year and to combine them with the corresponding snow cover information. Mean daily deposition of LAP on snow could have then been computed for each year of the period of interest and the inter-annual variability could have been estimated precisely. Second, ....To overcome these limitations, the MERRA2 atmospheric reanalysis (Gelaro et al., 2017) could be considered as an alternative to the GDFL dataset since MERRA2 provides global daily estimates of wet and dry LAP depositions rates at 50-km resolution since 1980. ....*

The values $D_{high}$ and $D_{low}$ for dust, BC and total LAP have been added to the dataset distributed on Zenodo (see version 2 of the corresponding dataset). This will provide the users of the climatology of LAP deposition on snow with an estimation of the interannual variability of LAP deposition on snow.

10. Considering just limited sites and poor spatial representativeness, upscaling from site to globe is not reliable. I don't think Figure 12 is accuracy enough. I suggest the authors remove the results related to global mapping. At least, the authors need to evaluate the global simulations with and without optimizing the parameters.

For a large number of processes, global models rely on parameterizations that have been calibrated at point scale and upscaled at large scale. We agree with Reviewer 1 that this upscaling always comes with uncertainties, and this is consequently also the case for the global map of gamma. One source of uncertainty is the interannual variability of LAP deposition on snow as discussed in our answer to the previous comment. We propose to account for this uncertainty in the revised paper by computing the values of gamma corresponding to a low LAP year and to a high LAP year using the regression given in Eq. 9 of the original paper and the map of LAP deposition on snow for a low LAP and high LAP year. Similarly to the coefficient of variability computed for the LAP deposition on snow, we propose to compute a coefficient of variability for the value of gamma:

$$C_{v,\gamma} = (\gamma_{high} - \gamma_{low})/\gamma_{mean}$$

Figure 12 in the submitted paper has been updated to add a map that shows how $C_{v,\gamma}$ varies globally (see Fig. 6 below). The results section has been adjusted as follows (L 463-468_:

*The largest uncertainties for gamma as shown on Fig. 13b are found in regions where the inter-annual variability of LAP deposition on snow is the largest (Fig. 7c). This includes the east coast of North America and central Siberia that are close to regions of BC emissions. The Karakum Desert, the Aral Sea and the Gobi Desert are also regions of large variability in gamma due to the strong inter-annual variability of dust deposition in these regions (Fig. 7b). Finally, Antarctica shows no inter-annual variability of gamma since the values of gamma for a low and high LAP in this region are equal to 900 days (the maximal value allowed for gamma in Crocus) due to very low LAP deposition on snow (Fig. 7c).*

[Figure]

*Figure 6: Global maps showing (a) the optimal value of γ derived and (b) the coefficient of variability of γ associated with the LAP inter-annual variability.*

The values $\gamma_{high}$ and $\gamma_{low}$ have been to the dataset distributed on Zenodo. This will allow the users of the model to know where the values of gamma are uncertain due to the interannual variability of LAP deposition on snow. Such uncertainty could be considered in the context of ensemble snowpack simulations.

Snowpack simulations with SVS2/Crocus with and without optimizing gamma and driven by ERA-5 have been recently produced for the paper describing the full SVS2/Crocus model in preparation for GMD. Results show strong improvements in terms of snow cover duration, in particular in the estimate of the snow melt-out date compared to satellite-derived estimates.

11. How does the model simulate the temporal evolution of LAP impacts? As I know, the LAP concentration in snow varies with time largely. Please show whether the optimized parameters improve the seasonal variations and interannual variability.

The default snow albedo parameterization in Crocus cannot capture the temporal evolution of LAP concentration at the surface of the snowpack and within the snowpack. The simulation of such processes with Crocus requires the use of the TARTE radiative transfer scheme (Libois et al., 2014) combined with the explicit simulation of the evolution of the LAP content (dust and BC) in the snowpack proposed by Tuzet et al. (2017). Only such an approach can be used to represent the impact of individual LAP deposition events that can strongly influence the evolution of the snow albedo (as mentioned L 449-451 in the submitted manuscript). In the default albedo parameterization of Crocus, the radiative impact of LAP on snow is taken into account via the snow darkening coefficient, gamma, which is fixed in time and linked to the mean LAPs deposition during the snow season. The strategy proposed in our paper aims at taking into account the spatial variability of LAP deposition on snow (from a climatological point of view) without additional numerical costs. Among the models used in the ESM-SnowMIP intercomparison (Menard et al., 2019), 19 out of 21 do not include any representation of the temporal and spatial variability of the impact of LAP on snow albedo. We believe the approach proposed in this study constitutes an improvement since it takes into account the effect of the spatial variability of LAP deposition on snow.

We explicitly mentioned in the discussion (L 547) and in the conclusion (L 597-599) of the revised paper that the method proposed in this study cannot capture the seasonal variation of LAP deposition on the snowpack.

*The methodology developed in this study relies on climatological values of LAP deposition on snow, so that it cannot represent the **seasonal variation of LAP deposition on snow** and the impact of individual LAP deposition events that can strongly influence the evolution of snow albedo (Di Mauro et al., 2015; Dumont et al., 2020).* (Discussion of the revised paper)

*This approach takes into account the climatological spatial variability of LAP deposition on snow from a  but cannot represent the seasonal variation of LAP deposition that can result from single deposition events as well as the interannual variability of LAP deposition.* (Conclusion of the revised paper)

**Minor concerns:**

1. Line 20-23: Topography can also affect snow albedo. We mentioned the impact of the topography in the introduction of the revised manuscript and referred to the paper by Picard et al. (2020).

*Picard, G., Dumont, M., Lamare, M., Tuzet, F., Larue, F., Pirazzini, R., & Arnaud, L. (2020). Spectral albedo measurements over snow-covered slopes: theory and slope effect corrections. The Cryosphere, 14(5), 1497-1517.*

2. Line 54: '…' _> etc. etc is used in the revised manuscript instead of '…'

3. Line 82: What is this variable? This semi-empirical variable is the sphericity of the snow grain. It varies between 0 and 1 and describes the ratio of angular versus rounded shape in a given snow layer. The text in the revised paper has been modified as follows (L 93-95):

*For each snow layer, Crocus simulates the evolution of the thickness, density, liquid water content, temperature, age, and snow microstructure represented by the snow specific surface area and the snow grain sphericity. The sphericity is a semi-empirical variable that describes the ratio of angular versus rounded shape in a given snow layer (Brun et al, 1992; Carmagnola et al, 2014).*

4. Figure 2: Please show the snow types in the figure. The snow classification from Sturm et al (2021) has been added as a background for Figure 2 in the revised manuscript. The revised figure is shown below:

[Figure]

5. Line 171: How did the authors calculate the daily albedo?

   The simulated daily snow albedo was computed as the average for a given day of the hourly snow albedo value simulated by the model. The information was added in the paper (L 206):

   *For the evaluation, the **hourly** simulated snow albedo was averaged over each day to retain an aggregated daily value similar to the observations.*

   A different treatment has been applied to the observations to get the daily values of observed snow albedo. Such treatment is required to remove the uncertainties associated with the measurements of incoming and outgoing SW radiation at the beginning and at the end of the day when the sun is low on the horizon. Such treatment is not required for the simulated snow

albedo. The paragraph describing the calculation fo the daily observed snow albedo has been revised as follows (L 176-184):

*To guarantee homogeneity between all ten sites, the observed daily-average albedo at the added sites was computed from hourly incoming and outgoing shortwave radiation using the same methods as for the ESM-SnowMIP sites (Morin et al., 2012; Ménard et al., 2019). Hourly values of incoming and outgoing radiation were filtered to discard hours when there was snowfall, when the incoming radiation was below 20 Wm-2 (to remove hours when the sun is low on the horizon), or when the outgoing radiation was below 2 Wm-2. For the five sites which were added in this study (Bylot, Umiujaq, Trail Valley Creek, Kühtai, and Sodankylä), another filter was added ensuring that the hourly incoming radiation was greater than the outgoing radiation. For all sites, the daily-averaged albedo was then computed by dividing the sum of hourly outgoing radiation values by the sum of hourly incoming radiation values. Days with less than 5 hours of valid radiation measurements were discarded.*

6. Please check Appendix A: Parameterizations used in Crocus. A sentence has been added at the beginning of Appendix A to better explain what is detailed in this part of the manuscript:

*"The multiphysics version of Crocus (Lafaysse et al., 2017) offers different options to simulate physical processes that drive the evolution of the snowpack. Table A1 details the options used in Crocus for all the simulations presented in this study."*

7. Equation 3-4: How about R square? The coefficient of determination (R**2) has not been considered in our study since it does not reflect systematics errors that are captured by the bias and RMSE considered in our study. The RMSE has been used to select the optimal range of gamma at each study site.

8. Figure 5: Is the site difference related to snow type? The differences in the range of gamma for each site shown on Figure 5 of the submitted paper do not depend on the snow type. They are mainly related to the differences of LAP deposition at these sites. As detailed above in our answer to the general comment 5, the direct influence of the snow type on the snow albedo in Crocus is taken into account via the snow optical diameter simulated by the snow metamorphism scheme.

**Answer to Reviewer 2**

We thank Reviewer 2 for their comments. We provide here our responses to those comments and describe how we addressed them in the revised manuscript. The original reviewer comments are in normal black font while our answers appear in blue font. The line numbers mentioned below correspond to the line numbers in the manuscript in track-change mode.

This research project aims to represent the effect of Light-Absorbing Particles (LAP) on snow albedo within the Crocus model as snow aging parameters, and to establish a global climatology of LAP deposition on snow, with the intention of applying these results to land surface models. The effort to achieve a global climatology of this phenomenon, by accounting for the altitude dependence of snow aging and making it regionally specific, is commendable. However, there has been insufficient discussion regarding the methodology for estimating the parameters related to snow aging and LAP climatology when scaling this approach globally. Consequently, it cannot be stated that the reliability of the climatology related to LAP deposition, which is the final outcome, is fully assured. In particular, the following points must be thoroughly discussed:

(major comments)

1. As the author demonstrates, the parameters associated with snow aging show significant differences between the accumulation and melting periods (Fig. 3). However, this study only considered the regional dependency of snow aging. It would be more beneficial for the authors to evaluate snow aging separately for the accumulation and melting periods and then discuss the relationship between snow aging and LAP deposition on the snow. In other words, the climatology of LAP deposition will be influenced not only by altitude but also by the timing of the accumulation and melting periods.

As mentioned by Reviewer 2, Figure 3 in the submitted manuscript shows large changes in the temporal evolution of the simulated albedo with contrasted behavior during the snow accumulation and melting periods. This evolution is mainly explained by four processes simulated by the model:

1. The albedo decrease due to the increase in snow optical grain diameter resulting from dry snow metamorphism during periods without precipitation and no melting. This evolution of the snow optical grain diameter is handled by the metamorphism scheme for dry snow in Crocus (Carmagnola et al., 2014).
2. The albedo decrease due to the increase in snow optical grain diameter resulting from wet snow metamorphism during melting periods. The evolution of the snow optical grain diameter is handled by the metamorphism scheme for wet snow in Crocus (Carmagnola et al., 2014).
3. The albedo decrease in the visible range due to LAP deposition. This effect is empirically represented by the value of gamma in Crocus in the albedo parameterization.
4. The albedo increase due to snowfall which is handled by the snowfall module in Crocus.

The three first processes can be considered as snow aging processes (snow metamorphism and LAP deposition) that are responsible for the decrease of snow albedo with time. However, gamma only directly impacts the third aging process. Therefore, we believe that the term "snow aging coefficient"

in the submitted paper was creating confusion for the reader. Reviewer 1 made a similar comment. For this reason, gamma has been renamed the "snow darkening coefficient" in the revised manuscript since this coefficient is only used to empirically represent the impact of LAP deposition on the snow albedo in the visible range. We believe this new name will help the reader to better understand the role of gamma in the evolution of snow albedo in Crocus. More explanations about the snow albedo parameterization in Crocus have been added to the revised manuscript (Section 2, L 108-112):

*The evolution of the optical grain size in Crocus is computed using metamorphism laws as described in Brun et al. (1992) and Carmagnola et al. (2014). For a given layer, for dry snow, the temporal evolution of the optical grain size is a function of the vertical temperature gradient. whereas, for wet snow, the increase in optical grain size with time depends on the snow liquid water content. New snow in Crocus is characterized by an optical diameter of 1e-4 m (surface specific area of 65 m2/kg) and results in an increase in snow albedo.*

For the reasons mentioned above, we believe that the snow darkening coefficient should not be too different during the snow accumulation and melting periods since it is not associated with the physical processes that drive the large differences of snow albedo evolution during the accumulation and melting seasons. For example, the impact of the presence of liquid water on the snow optical grain size is handled by the metamorphism scheme in Crocus. Instead, gamma can be seen as an average quantity of LAP in the snowpack. The work proposed in this study allows spatially varying values of gamma based on the climatology of LAP deposition on the snowpack. However, it cannot present the interannual and seasonal variability of LAP deposition on the snowpack. It is now mentioned in the discussion (L 547) and conclusion (L 597-599) of the revised paper:

*The methodology developed in this study relies on climatological values of LAP deposition on snow, so that it cannot represent the **seasonal variation of LAP deposition on snow** and the impact of individual LAP deposition events that can strongly influence the evolution of snow albedo (Di Mauro et al., 2015; Dumont et al., 2020).* (Discussion of the revised paper)

*This approach takes into account the climatological spatial variability of LAP deposition on snow but cannot represent the seasonal variation of LAP deposition that can result from single deposition events as well as the interannual variability of LAP deposition.* (Conclusion of the revised paper)

2. (L173) If the optical thickness of the snow cover is insufficient, the snow albedo will be influenced by the albedo (reflectance) of the ground surface beneath the snow cover. Consequently, γ (gamma) will also reflect the influence of the ground surface. In forested areas, the effect of vegetation must be considered as well. Therefore, γ is not only affected by snow aging but may also be strongly influenced by local factors. It is necessary to provide a comprehensive explanation of this point when determining the relationship between D (climatological deposition rate of LAPs) and γ.

As mentioned in our answer to the first point raised by Reviewer 2, the snow darkening coefficient gamma only represents the effect of LAP deposition on snow albedo in the visible band in Crocus. To avoid the effect of ground contamination on the measurement of snow albedo, a threshold on snow depth has been applied to remove periods when the snow cover was not thick enough. A threshold value of 20 cm was selected for the sites below 60 N and 10 cm was used for the sites

above 60 N (L 175-177 of the submitted manuscript). Snow albedo calculations using the online version of the snow radiative transfer model SNICAR v3 (Flanner et al., 2021) (SNICAR-Online (umich.edu)) have been used to quantify the impact of ground contamination on broadband snow albedo for snow depth values corresponding to the threshold values selected in our study. Two snowpack have been considered: (i) a snowpack made of fresh snow with high SSA and low density values (typical of the accumulation season) and (ii) a snowpack made of melt forms) with low SSA and high density values (typical of the ablation season. The SSA and density values for each type of snow were taken from Domine et al. (2007). The results are presented in Table 1 below. For a 10-cm (20-cm) thick snowpack made of fresh snow, the ground contamination reduces the snow albedo by 0.8 % (0.2 %). For a 10-cm (20-cm) thick snowpack made of melt forms, the ground contamination reduces the snow albedo by 1.9 % (0.6 %).

Table 1: Snow albedo computed by SNICAR for 2 different snowpack of various thickness. .

| | Snow albedo values for different snowpack thickness | | |
|---|---|---|---|
| Type of snowpack and ground | Thickness = 0.1 m | Thickness = 0.2 m | Thickness = 100 m (optically infinite) |
| SSA: 65 m2/kg
Density: 150 kg/m3
Ground albedo: 0.25
No LAP | 0.841090 | 0.846047 | 0.848279 |
| SSA: 10 m2/kg
Density: 300 kg/m3
Ground albedo 0.25
No LAP | 0.772633 | 0.782954 | 0.787879 |

Based on this analysis, we can consider that the threshold values on snow depth applied in this study are sufficient to limit the impact of ground contamination on the snow albedo measurements. This allows us to have a robust comparison between simulated and observed snow albedo and to reduce the potential impact of ground contamination on the optimal ranges of gamma derived at each site. The results of the simulation with SNICAR were added to the supplementary material (Section S2.2). We also mentioned these results in the revised paper when describing the criteria imposed to select days for the albedo evaluation (L 209-216):

*The first criterion ensured that the observed albedo corresponded to an actual snow albedo value. It made sure that the albedo measurement had been collected over a fully snow-covered ground and that the ground surface beneath the snow cover did not affect the measurement. Observed SD had to be higher than 20 cm. For high-latitude sites (above 60°N), this criterion was relaxed to 10 cm because there is less precipitation at these sites, so SD did not reach far over 20 cm in certain years. Albedo simulations with the radiative transfer model SNICAR v3 (Flanner et al., 2021) confirmed the*

*ground surface beneath the snow cover does not modify the snow albedo by more than 2 % for a 10-cm thick snowpack (see Section S2.2 of the supplementary material)*

We fully agree with Reviewer 2 that forest debris from surrounding high vegetation can influence the evolution of snow albedo in the visible range in forested terrain as shown by Melloh et al. (2001). Consequently, gamma should be modified in presence of high vegetation to represent this effect. The present study focuses only on the large-scale variability of the snow darkening coefficient in open conditions (without trees) as a function of LAP deposition (BC and dust). For this reason, the three forested sites available in the ESM-SnowMIP dataset were not used in this study due to the potential impact of forest debris that could influence the optimal ranges of gamma at these sites as mentioned at L 136-138 in the submitted manuscript. The impact of forest debris on the values of gamma has been included in a new paragraph of the discussion (L 569-573):

*In addition, the effects of forest litter and debris on snow albedo in the visible range (Melloh et al., 2001) and on the resulting gamma values were not considered in this study. Only sites located in open terrain were selected to determine the optimal ranges of gamma. Therefore, the values of gamma shown on Fig. 12 do not include the effect of the forest presence. The approach of Hardy et al. (2000) could be considered to indirectly represent the effects of forest litter in the default snow albedo scheme used in Crocus.*

3. The climatology of LAP deposition will be a valuable dataset for global land surface models. However, the results presented in Fig. 12 have not been fully validated, and their accuracy, including associated uncertainties, remains unclear. Additionally, because the validation sites are limited, there are likely to be high uncertainties in the LAP data for regions such as South Asia, particularly in mountain glaciers. The authors should release the dataset only after thorough validation.

We agree with Reviewer 2 that the climatology of LAP deposition on snow and the associated values of the snow darkening coefficient are associated with uncertainties. To answer this comment, two additions were made to the revised paper:

1. An evaluation of the quality of the GFDL LAP deposition dataset
2. A quantification of the uncertainty in the values of the snow darkening coefficient resulting from the interannual variability of LAP deposition on snow.

1. A special section describing the accuracy of the GFDL LAP product has been added to the Supplementary material (Section 1.2). It contains a comparison between observed and simulated mean annual dust deposition at 26 sites around the world as shown in Figure 1 below. The GFDL product captures well the spatial variability of dust deposition around the globe. For example, it reproduces well, the contrasted dust deposition patterns in Central Asia (Fig 1b). For this evaluation, values from 6 different mountain glaciers in Asia have been considered (Table 1). The evaluation of BC climatological deposition is restricted to three sites at the moment as shown in Table 3. It shows that the LAP dataset captures the large differences in BC deposition between the Himalayan mountains and West Antarctica.

[Figure]

Figure 1 (a) Map showing the mean annual dust deposition from the GFDL climatology (1981-2015), (b) same as (a) for a region centered around the Tibetan Plateau and (c) scatter plot comparing the annual dust deposition in the observation and in the GDFL climatology at 26 sites around the globe. The location of these sites is shown on maps (a) and (b). For each site, the pie chart shows the decomposition in the GFDL climatology between dry (orange) and wet (blue) deposition.

*Table 2  Mean annual dust deposition in the observations and in the GFDL climatology at 26 sites. The numbers correspond to the numbers shown on Fig 1.*

| Num. | Name | Lat. | Lon. | Elev. (m) | Reference | Obs. ($g\,m^{-2}yr^{-1}$) | Clim. ($g\,m^{-2}yr^{-1}$) |
|---|---|---|---|---|---|---|---|
| 1 | Taklimakan | 39.75°N | 88.50°E | 200. | Zhang et al. (1998) | 450.00 | 177.37 |
| 2 | Waddan | 29.12°N | 15.93°E | 2. | O'Hara et al. (2006) | 74.53 | 34.61 |
| 3 | Tel Aviv | 32.00°N | 34.50°E | 40. | Ganor and Mamane (1982) | 36.00 | 26.87 |
| 4 | Ghana, Gulf of Guinea | 7.50°N | -1.50°E | 0. | Resch et al. (2008) | 22.00 | 9.66 |
| 5 | Erdemli | 33.57°N | 34.26°E | 21. | Kubilay et al. (2000) | 13.00 | 9.99 |
| 6 | Mera Ice Peak | 27.72°N | 86.87°E | 6376. | Ginot et al. (2014) | 10.40 | 11.39 |
| 7 | Karakoram Mtns | 36.00°N | 75.70°E | 5150. | Wake et al. (1994) | 9.60 | 11.28 |
| 8 | Pamir Mtns | 38.20°N | 75.10°E | 5910. | Wake et al. (1994) | 8.14 | 20.27 |
| 9 | Tuomuer Glacier | 41.75°N | 79.87°E | 4600. | Zhiwen and Zhongqin (2011) | 8.00 | 31.69 |
| 10 | Montseny Mtns | 41.80°N | 2.30°E | 700. | Avila et al. (1997) | 5.30 | 4.64 |
| 11 | So. Tibetan Plateau | 28.50°N | 87.50°E | 5850./6140. | Wake et al. (1994) | 3.39 | 5.28 |
| 12 | Tanggula Shan Mtns | 33.40°N | 91.10°E | 5950. | Wake et al. (1994) | 3.02 | 2.81 |
| 13 | French Alps | 45.50°N | 6.50°E | 4270. | Angelisi and Gaudichet (1991) | 2.10 | 3.81 |
| 14 | Miami | 25.75°N | -80.25°E | 10. | Prospero et al. (1987) | 1.62 | 1.10 |
| 15 | Tasman Glacier | -43.50°N | 170.30°E | 2600. | Windom (1969) | 1.20 | 0.82 |
| 16 | Renland | 71.30°N | -26.70°E | 2340. | Bory et al. (2003) | 0.68 | 0.73 |
| 17 | Midway | 28.20°N | -177.35°E | 4. | Prospero (1989) | 0.60 | 0.28 |
| 18 | Shemya | 59.92°N | 174.00°E | 2. | Prospero (1989) | 0.60 | 0.65 |
| 19 | Navarino Island | -55.22°N | -67.62°E | 35. | Sapkota et al. (2007) | 0.43 | 0.53 |
| 20 | Oahu | 21.30°N | -157.60°E | 20. | Prospero (1989) | 0.42 | 0.13 |
| 21 | Mount Olympus | 47.00°N | -123.00°E | 2000. | Windom (1969) | 0.32 | 0.47 |
| 22 | New Zealand | -34.50°N | 172.75°E | 2. | Arimoto et al. (1990) | 0.14 | 0.41 |
| 23 | Fanning | 3.90°N | -159.30°E | 25. | Prospero (1989) | 0.05 | 0.04 |
| 24 | James Ross Island | -64.20°N | -57.70°E | 1542. | McConnell et al. (2007) | 0.03 | 0.20 |
| 25 | GRIP | 72.60°N | -37.60°E | 3232. | Bory et al. (2003) | 0.02 | 0.18 |
| 26 | Enewetak | 11.30°N | 162.30°E | 5. | Arimoto et al. (1985) | 0.02 | 0.04 |

*Table 3: Mean annual black carbon deposition in the observations and in the GFDL climatology at three sites.*

| Num. | Name | Lat. | Lon. | Elev. (m) | Reference | Obs. ($g\,m^{-2}yr^{-1}$) | Clim. ($g\,m^{-2}yr^{-1}$) |
|---|---|---|---|---|---|---|---|
| 1 | Nepal Clim. Obs.- Pyramid | 27.95°N | 86.80°E | 5079. | Yasunari et al. (2010) | 112.00 | 179.83 |
| 2 | Tibet Palong-Zanbu Glacier | 29.21°N | 96.92°E | 5500. | Xu et al. (2009) | 29.40 | 77.40 |
| 3 | West Antarctic Ice Sheet | -79.46°N | -112.08°E | 1766. | Bisiaux et al. (2012) | 0.02 | 0.07 |

2. The impact of the interannual variability of LAP deposition is now considered in the revised version of the manuscript. The GFLD dataset consists of different percentiles (1, 5, 10, 25, 50, 75, 90, 95, 99) of BC and dust deposition rates over the period 1979-2015, for each day of the year. In the submitted paper, the median deposition rate for each day was considered and combined with the GMASI daily snow cover climatology to obtain an estimation of the mean daily LAP deposition on snow. In the

revised manuscript, we have considered the percentiles 25 and 75 for dust and BC to obtain an estimation of typical daily average LAP deposition on snow for a low LAP year and for a high LAP year. This method provides an estimation of the interannual variability of LAP deposition on snow.

The daily average deposition of dust and BC on snow for a low (high) LAP year has been computed by replacing the median deposition rate by the percentile 25 (75) for each day of the year in Eq. 5 of the submitted paper. The daily average deposition of LAP on snow for a low (high) LAP has been finally expressed in equivalent BC using Eq. 6 of the submitted paper. A coefficient of variability, $C_{v,D}$, has been computed to characterize the inter-annual variability of LAP deposition on snow with respect to the climatology:

$$C_{v,D} = (D_{high} - D_{low})/D_{mean}$$

Where $D_{high}$, $D_{low}$ and $D_{mean}$ corresponds to the daily average LAP deposition on snow for a high LAP year, a low LAP year and the climatology. Figure 2 and 3 below show how the interannual range LAP deposition on snow ($D_{high} - D_{low}$) and the coefficient of variability, $C_{v,D}$, vary in space. Figure 2 has been added to the supplementary material of the revised paper (Figure S.3) whereas Figure 3 has been added to the main revised manuscript (Figure 8 in the revised document)

[Figure]

Figure 2: Global maps of the interannual range of daily (a) BC, (b) dust, and (c) total LAP (BC + dust in equivalent BC) deposition rates over snow between a high and a low LAP year ($D_{high} - D_{low}$).

[Figure]

*Figure 3: Global maps of the coefficient of variability of annual mean (a) BC, (b) dust, and (c) total LAP (BC + dust in equivalent BC) deposition rates over snow between a high and a low LAP year.*

A new paragraph has been added to the results section to describe Figure 3 (L 399-407):

*Figure 8 quantifies the inter-annual variability of BC, dust and total LAP deposition on snow. The Mediterranean basin, Turkey, the Caucasus, the southern Andes and the mountains of Western US presents a large inter-annual variability of dust deposition on snow associated with the strong variability of dust deposition events in these regions (Skiles et al., 2012; Di Mauro et al., 2015; Dumont et al., 2020; Reveillet et al., 2022; Haugvaldstad et al., 2024) . The inter-annual variability of dust deposition on snow is generally larger than the inter-annual variability of BC deposition on snow, in agreement with (Reveillet et al., 2022). Antarctica appears as the region with the strongest inter-annual variability of BC deposition on snow (compared to the climatology) but this variability remains the lowest around the planet when considered in absolute value (Fig. S4 in the Supplementary Material). As for the total deposition rates, the inter-annual variability of total LAP deposition on snow remains close to the values found for BC, at the exception of regions where the inter-annual variability of dust deposition on snow is the largest (Karakum Desert, Aral Sea, Gobi Desert and south-western Andes).*

The LAP deposition rates for a low and a high LAP year are shown in the cross analysis to illustrate the interannual variability of LAP deposition of snow at each of the experimental sites considered in the study. Figure 9 in the submitted manuscript has been updated to show the interannual variability at all the different sites and the effect of the altitudinal correction (see Fig. 4 below for the revised figure). This figure shows that, despite the interannual variability, sites such as Sapporo or Col de Porte receive significantly larger amounts of LAP on snow than Arctic sites such as Bylot and Umiujaq. The regression between the optimal values of gamma and the climatological LAP deposition on snow at each of the sites were not modified in the revised manuscript. Indeed, the optimal ranges of gamma at a given site have been selected based on the median RMSE of simulated snow albedo over all the years for each value of gamma. This range is therefore related to the mean climatological LAP deposition on snow at each of the sites. Figure 10 in the submitted manuscript showing the linear regression between gamma and LAP has also been updated to show the interannual variability of LAP deposition on snow at each site (see Fig. 5 below for the revised figure).

[Figure]

*Figure 4: Graph representing climatological LAP deposition rates at all sites, before and after the correction of elevation effects. Arrows indicate these corrections at mountainous sites. The error bar represents the interannual variability of LAP deposition at each site.*

[Figure]

*Figure 5: Graph representing the optimal ranges of γ as a function of total LAP deposition rates in equivalent BC, in logarithmic scales, and the corresponding linear regressions with and without correction for elevation at the mountain sites (the correction is represented by the arrows). The error bar represents the interannual variability of LAP deposition at each site.*

The estimation of the interannual variability of LAP deposition on snow is calculated for a low LAP year and a high LAP year using the percentiles 25 and 75 for dust and BC deposition available in the GFDL dataset. The limitations associated with this approach are mentioned in the discussion (L509-525):

*The LAP deposition outputs in the GFDL dataset only contain the deposition rates for BC and dust as percentiles for each day of the year, based on the 1979-2015 period. This characteristic of the dataset led to two limitations of this study. First, the inter-annual variability of LAP deposition on snow was calculated for a low LAP year and a high LAP year using the percentiles 25 and 75 for dust and BC deposition for each day of the year. Such a method can only provide an estimation of the inter-annual variability of LAP deposition. A more accurate computation would have required to use the deposition rates of BC and dust for each day of the year and to combine them with the corresponding snow cover information. Mean daily deposition of LAP on snow could have then been computed for each year of the period of interest and the inter-annual variability could have been estimated precisely. Second, ....To overcome these limitations, the MERRA2 atmospheric reanalysis (Gelaro et al., 2017) could be considered as an alternative to the GDFL dataset since MERRA2 provides global daily estimates of wet and dry LAP depositions rates at 50-km resolution since 1980. ....*

The values $D_{high}$ and $D_{low}$ for dust, BC and total LAP have been added to the dataset distributed on Zenodo (see version 2 of the corresponding dataset). This will provide the users of the climatology of LAP deposition on snow with an estimation of the interannual variability of LAP deposition on snow.

The uncertainty in the estimation of gamma resulting from the interannual variability of LAP deposition on snow has been quantified in the revised paper. We proposed to account for this uncertainty in the revised paper by computing the values of gamma corresponding to a low LAP year and to a high LAP year using the regression given in Eq. 9 of the original paper and the map of LAP deposition on snow for a low LAP and high LAP year. Similarly to the coefficient of variability computed for the LAP deposition on snow, we propose to compute a coefficient of variability for the value of gamma:

$$C_{v,\gamma} = (\gamma_{high} - \gamma_{low})/\gamma_{mean}$$

Figure 12 in the submitted paper has been updated to add a map that shows how $C_{v,\gamma}$ varies globally (see Fig. 6 below). The results section has been adjusted as follows (L 463-468):

*The largest uncertainties for gamma as shown on Fig. 13b are found in regions where the inter-annual variability of LAP deposition on snow is the largest (Fig. 7c). This includes the east coast of North America and central Siberia that are close to regions of BC emissions. The Karakum Desert, the Aral Sea and the Gobi Desert are also regions of large variability in gamma due to the strong inter-annual variability of dust deposition in these regions (Fig. 7b). Finally, Antarctica shows no inter-annual variability of gamma since the values of gamma for a low and high LAP in this region are equal to 900 days (the maximal value allowed for gamma in Crocus) due to very low LAP deposition on snow (Fig. 7c).*

[Figure]

*Figure 6: Global maps showing (a) the optimal value of γ derived and (b) the coefficient of variability of γ associated with the LAP inter-annual variability.*

The values $\gamma_{high}$ and $\gamma_{low}$ have been to the dataset distributed on Zenodo. This will allow the users of the model to know where the values of gamma are uncertain due to the interannual variability of LAP deposition on snow. Such uncertainty could be considered in the context of ensemble snowpack simulations.

(minor comments)

1. It is understood that Crocus is a multi-layer model used to simulate snow conditions across various layers. However, only two snow grain size parameters (d_opt and d') are employed to calculate the albedo in the visible and near-infrared regions (Table 1). In the visible range, ice exhibits weak light absorption, allowing light to penetrate more deeply. Therefore, to accurately calculate the albedo in the visible range, it is crucial to consider the vertical distribution of snow grain sizes. Please add an explanation of which snowpack layer d_opt parameter represents.

Thanks for this comment. We agree with Reviewer 2 that the description of how Crocus handles solar radiation was not sufficient in the submitted manuscript. We have added the following information about the model. First, the snow albedo in Crocus is computed as a depth-weighted average of the albedo of the two upper snow layers, so on a total depth of a few centimeters. The properties (d_opt, age) of each layer are used to compute their respective albedo. This method is applied to avoid time discontinuities in the simulated snow albedo resulting from layer aggregation. Second, Crocus simulates the penetration of incoming shortwave radiation into the snowpack and its absorption assuming an exponential decay of radiation with increasing snow depth. The absorption coefficient in the different spectral bands depends on the optical diameter and on the density of each snow layer.

The following sentences were added to the revised paper:

*The snow albedo in each band is obtained as a depth-weighted average of the albedo of the two upper snow layers. The properties (d_opt, age) of each layer are used to compute their respective albedo. This method is applied to avoid time discontinuities in the simulated snow albedo resulting from layer aggregation. (L112-115)*

*Crocus simulates the penetration of incoming shortwave radiation into the snowpack and its absorption assuming an exponential decay of radiation with increasing snow depth. The absorption coefficient in the different spectral bands (Table 1) depends on the optical diameter and on the density of each snow layer. (L120-122)*

Table 1 has been revised to include the snow absorption coefficient for each band:

**Table 1.** Equations representing the snow albedo and the snow absorption coefficient for the three spectral bands in Crocus. The parameters are as follows: $d_{opt}$ (m) is the optical grain diameter of the snow, $\rho$ the snow density, $P$ (Pa) is the mean pressure at the site, $P_{CDP}$ (Pa) is the mean pressure at the Col de Porte site, $A$ (days) is the age of the snow, and $\gamma$ (days) is the snow  darkening coefficient. Adapted from Table 4 in Vionnet et al. (2012).

| Spectral band | Spectral albedo $\alpha_{spectral\,band}$ | Absorption coefficient $\beta$ $(m^{-1})$ |
|---|---|---|
| 0.3—0.8 μm | $\alpha_{0.3-0.8\mu m} = \max(0.6, \alpha_i - \Delta\alpha_{age})$
 $where: \alpha_i = \min(0.92, 0.96 - 1.58\sqrt{d_{opt}})$
 $and: \Delta\alpha_{age} = \min(1, \max(\frac{P}{P_{CDP}}, 0.5)) \times 0.2\frac{A}{\gamma}$ | $\beta_{0.3-0.8\mu m} = \max(40, 0.00192\frac{\rho}{\sqrt{d_{opt}}})$ |
| 0.8—1.5 μm | $\alpha_{0.8-1.5\mu m} = \max(0.3, 0.9 - 15.4\sqrt{d_{opt}})$ | $\beta_{0.8-1.5\mu m} = \max(100, 0.01098\frac{\rho}{\sqrt{d_{opt}}})$ |
| 1.5—2.8 μm | $\alpha_{1.5-2.8\mu m} = 346.3d' - 32.31\sqrt{d'} + 0.88$
 $where: d' = \min(d_{opt}, 0.0023)$ | $\beta_{1.5-2.8\mu m} = +\infty$ |

2. Please explain how the solar zenith angle dependence of snow albedo and the effects of the atmosphere, especially clouds, on snow albedo can be expressed in terms of Eq. 1.

The dependence of snow albedo on the solar zenith angle and the influence of the cloud cover and optical thickness on the distribution of the incoming radiation in the three spectral bands is not taken into account in the default albedo scheme in Crocus. This limitation is explicitly mentioned in the revised manuscript when detailing the snow albedo parameterization in Crocus (L 134-136):

*The model does not include the change in the ratio of incoming radiation in the three bands as a function of the sky conditions and of the solar zenith angle. The impact of these limitations is discussed in Sect. 5.*

We then discuss the uncertainties associated with these limitations in the discussion of the revised paper (L 541-545):

*Despite the improvements brought in this study, the default snow albedo in Crocus still suffers from limitations associated with the fixed ratio used to compute the broadband albedo from the values in the three spectral bands (Eq. 1) and the absence of effect of the solar zenith angle on snow albedo. Gardner and Sharp (2010) have shown that changes in solar zenith angle and clouds' optical thickness can lead to changes on the order of 0.05 in the snow albedo values (Fig. 9 in their study). Including these effects will lead to further improvements in large scale snow albedo simulations with Crocus.*

3. For reference, please also show the results for SSA=10 m^2kg^-1 (~granular snow) in Fig. 1.

The two figures below show how the snow albedo decrease with snow age varies as a function of the snow darkening coefficient gamma for two values of SSA (50 m$^2$ kg$^{-1}$ as in the submitted paper and 10 m$^2$ kg$^{-1}$ as suggested by Reviewer 2). In the visible range (Figure 2), the two graphics are identical since the reference albedo value for a snow age of 0 is equal to 0.92 for both values of SSA (see Table 1 in the main manuscript). On the other hand, the two SSA values lead to different albedo values in

the two near-infrared bands so that for a given age and a given value of gamma, the broadband albedo is lower for SSA = 10 m² kg⁻¹ than for SSA = 50 m² kg⁻¹.

We have decided to keep unchanged Figure 1 in the revised manuscript since the two values of SSA give the same albedo evolution as a function of snow age in the visible range.

[Figure]

*Figure 7: Graphic representation of the dependency of the snow albedo in the visible spectral band (0.3−0.8 µm) to the snow age for different values of gamma and for two SSA values (50 m2/kg and 10 m2/kg)*

[Figure]

*Figure 8: Graphic representation of the dependency of the broadband snow albedo (0.3−0.8µm) to the snow age for different values of gamma and for two SSA values (50 m2/kg and 10 m2/kg)*

---

## Author Response (AR2)

**Answer to minor revisions.**

Dear Editor,

Thanks for handling the review process of our paper submitted to TC. We have considered all the comments made during the second stage of reviews and changed the manuscript accordingly. The changes are described below in blue font and can be found in the manuscript in track-change mode.

All the best,

Vincent Vionnet

**Comments of Reviewer 1**

1. Please use large font size for Figure 7, 8, 13.

The font size of the color bars has been modified for the three figures.

2. Please improve the layout of Figure 6.

As suggested by the editor, the space between the main figure and the color bar has been reduced. In addition, the title and the legend of the figure have been revised to be more consistent.

3. In the abstract, the authors mentioned that the revised methods improved snow albedo simulations on average by 10% with the largest improvements found in the Arctic (more than 25%). Please indicate which metric was used to evaluate the model performance, and which benchmark dataset was used.

The abstract has been modified as follows:

*The revised coefficient improved snow albedo simulations at the ten experimental sites (average reduction in root-mean-square error, RMSE, of 10%) with the largest improvements found for the sites in the Arctic (RMSE reduced by 25%).*

**Comments of the Editor**

- L. 113: The mathematical symbol "dopt" is not defined in the running text. It is better to indicate the mathematical symbol and its definition together here, like "(optical grain diameter dopt, age A)."

"dopt" is now defined in the revised manuscript following the recommendation made by the editor.

- L. 130, L. 329, L. 460: "gamma" -> "γ"

The Greek letter γ is now used in the text.

- L. 393: "inter-annual variability": It is helpful for readers to add "(Eq. 7)" after this part.

The reference to Eq. 7 has been added to the text.

**Comments of the editorial office**

1. Please note that the title in your *.pdf manuscript should not be all capitalized.

The title in the revised manuscript read as:

*Improving large-scale snow albedo modeling using a climatology of light-absorbing particle deposition*

2. For the next revision please remove the placeholder "TEXT" from your manuscript or fulfill it with the information.

The placeholder "TEXT" has been removed from the revised manuscript